# Rho-ROCK liberates sequestered claudin for rapid *de novo* tight junction formation

**Yuma Cho**[1], **Akari Taniguchi**[1], **Akiharu Kubo**[2], **Junichi Ikenouchi**[1]*

[1]Department of Biochemistry, Kyushu University Graduate School of Medical Sciences, Fukuoka, Japan; [2]Division of Dermatology, Department of Internal Related, Graduate School of Medicine, Kobe University, Kobe, Japan

## eLife Assessment

This paper identifies a crucial step in the regulation of tight junction formation by identifying Rho-ROCK activity-dependent activation of the serine protease Matriptase, making Claudins available for tight junction formation. The reviewers were satisfied with the revisions and found the work **important** and the approach **convincing**.

## Abstract

The epithelial cell sheet maintains its integrity as a barrier while undergoing turnover of constituent cells. To sustain the barrier continuously, it's essential to preserve the 'old' tight junctions (TJs) between cells being excluded from the sheet and their neighbors while simultaneously forming de novo TJs between newly adjacent cells. However, the molecular mechanisms involved in the formation of de novo TJs remain largely unknown. This study investigates two scenarios: the formation of de novo TJs during the removal of apoptotic cells from mouse monolayer epithelial sheets and during the differentiation of the granular layer in mouse stratified epidermis. We revealed that rapid claudin assembly is achieved by actively regulating the dissociation of the EpCAM/TROP2-claudin complex in both situations. Furthermore, we found that the Rho-ROCK pathway initiates the activation of matriptase, which cleaves EpCAM/TROP2, resulting in the supply of polymerizable claudin from the stockpiled EpCAM/TROP2-claudin complex at the plasma membrane to induce rapid de novo TJ formation.

**\*For correspondence:**
ikenouchi.junichi.033@m.kyushu-u.ac.jp

## Introduction

The surfaces of our body and its organs are covered by epithelial cell sheets, which serve to prevent the entry of antigens and pathogens from the outside and the leakage of substances such as water and glucose from the body (*Piontek et al., 2020*; *Horowitz et al., 2023*). This barrier function is achieved through the close attachment of the plasma membranes of neighboring epithelial cells by intercellular adhesion structures called TJs. The major component of TJs is claudin, a four-transmembrane protein of around 25 kDa. Claudin binds in cis on the plasma membrane and in trans between neighboring cells, forming a membrane structure called TJ strands (*Tsukita et al., 2001*).

In both simple epithelia, such as the gastrointestinal tract, and stratified epithelia, such as skin, the barrier is continuously maintained by various mechanisms that regulate the homeostasis of the epithelial cell sheet, including the wound healing response (*Gu and Rosenblatt, 2012*). For example, unnecessary inflammatory responses are suppressed by mechanisms that rapidly eliminate apoptotic cells generated by constant cell turnover or exposure to external stresses in the epithelial cell sheet (*Duszyc et al., 2023*). During the process of apoptotic cell extrusion, the apoptotic cells are physically pushed towards the apical side by an increase in actomyosin contractility of cells adjacent to the apoptotic cells (*Takeuchi et al., 2020*; *Rosenblatt et al., 2001*). The barrier function of the epithelial

sheet is maintained during this process of apical extrusion (*Rosenblatt et al., 2001*). In parallel with rapid apoptotic cell elimination, it is necessary to maintain the integrity of TJs between apoptotic cells and neighboring cells, while rapidly forming de novo TJs between newly adjacent cells that result from apoptotic cell elimination. However, the mechanism for the formation of such rapid de novo TJs is largely unknown.

In the epidermis, TJs form only between keratinocytes in the granular layer and act as a barrier separating the inside and outside of the body. Claudin-1 knockout mice die within a day of birth due to abnormal stratum corneum differentiation and disrupted granular layer barrier function, resulting in severe dehydration (*Furuse et al., 2002*; *Sugawara et al., 2013*). In the epidermis, stem cells in the basal layer divide to give rise to new keratinocytes, which differentiate towards the body surface into keratinocytes constituting the spinous layer, granular layer, and stratum corneum, forming a multilayered epithelium (*Kulukian and Fuchs, 2013*). TJs are only formed in the second layer of the granular layer (SG2), even though claudin is also expressed in cells of the basal and spinous layers (*Brandner et al., 2002*; *Yoshida et al., 2013*; *Yokouchi et al., 2016*). In the turnover of keratinocytes in the SG2 layer, the epidermal barrier is continuously maintained by the simultaneous formation of 'new' TJs between newly differentiating SG2 keratinocytes and pre-existing SG2 keratinocytes, before the 'old' TJs between SG2 keratinocytes that are differentiating into keratinocytes of the stratum corneum and neighboring SG2 keratinocytes are lost (*Yokouchi et al., 2016*). It has recently been reported that an E-cadherin-dependent increase of intercellular tension at adherens junctions (AJs) and inhibition of EGFR activity are required for induction of TJ formation in the SG2 layer (*Rübsam et al., 2017*). However, how these E-cadherin-mediated changes regulate TJ formation in the SG2 layer and why TJs are never formed in keratinocytes other than the SG2 layer that express claudin remain open questions.

The formation of de novo tight junctions, which is common in these two phenomena, is rapid, suggesting that the formation of functional TJs is not regulated through de novo transcription or translation of claudin, but rather by controlling the polymerization state of claudin. In heterogeneous cells, such as L cells and COS-7 cells, forced expression of claudin alone is known to be sufficient to induce TJ formation (*Furuse et al., 1998*; *Gonschior et al., 2022*). On the other hand, in the lateral membranes of simple epithelial cells and keratinocytes, excess claudin is present in an unpolymerized. The molecular mechanisms that strictly regulate the polymerization of claudin at the presumptive TJ region are largely unknown.

Recently, a single transmembrane protein, EpCAM, was identified as an interacting protein of claudin (*Ladwein et al., 2005*). EpCAM binds to claudin in the lateral membrane of epithelial cells through the interactions between transmembrane regions (*Nübel et al., 2009*; *Barth et al., 2018*). By binding to EpCAM, claudin is stabilized at the plasma membrane, preventing it from being endocytosed and degraded (*Wu et al., 2013*). Knockout of EpCAM in cultured cells does not affect claudin accumulation at TJs but selectively reduces claudin in the lateral membrane (*Wu et al., 2013*; *Higashi et al., 2023*).

Another single transmembrane protein, TROP2, which shows 67% similarity in amino acid sequence to EpCAM, also binds to claudin (*Lenárt et al., 2020*). In the small intestine, only EpCAM is expressed, while in the epidermis, both EpCAM and TROP2 are expressed (*Nakato et al., 2020*; *Szabo et al., 2022*). In humans, loss-of-function mutations in the EpCAM gene cause a disruption in the barrier function of gastrointestinal epithelial cells, resulting in congenital tufted enteropathy (CTE), characterized by chronic diarrhea and growth delay (*Sivagnanam et al., 2008*; *Kozan et al., 2015*). EpCAM knockout mice also show similar phenotypes and die within two weeks after birth (*Guerra et al., 2012*; *Lei et al., 2012*). Similarly, TACSTD2, which encodes the TROP2 protein, is the causative gene for human gelatinous drop-like corneal dystrophy (GDLD), which causes photophobia and vision loss due to amyloid deposition under the corneal epithelium (*Tsujikawa et al., 1999*; *Takaoka et al., 2007*). This is because the barrier function of the corneal epithelium is impaired due to TROP2 dysfunction, allowing lactoferrin and apolipoproteins in the tear fluid to penetrate and form amyloid deposits. It was also recently shown that the epidermal barrier function is impaired in TROP2 KO mice (*Szabo et al., 2022*).

EpCAM and TROP2 are cleaved in their extracellular regions by matriptase, a transmembrane serine protease (*Wu et al., 2017*; *Wu et al., 2020*). Cleavage of EpCAM disrupts the EpCAM-claudin complex, making available unpolymerized claudin for TJ formation (*Higashi et al., 2023*; *Wu et al.,*

*2020*). Higashi et al. proposed the physiological significance of this molecular mechanism, suggesting that when TJs are locally disrupted, serine proteases on the apical side shift their localization to the basolateral side and cleave EpCAM at the basolateral membrane, releasing unpolymerized claudin to repair TJs. In other words, local disruption of TJs allows passive translocation of proteases due to the partial loss of the fence function of TJs, leading to the breakdown of the EpCAM-claudin complex. In this study, we examined in detail the regulation of the EpCAM-claudin complex in the mechanisms to induce de novo TJs during the elimination of apoptotic cells and turnover of the SG2 layer of the epidermis.

## Results

### Rapid de novo TJ formation occurs during apoptotic cell elimination in a Rho-ROCK pathway-dependent manner

The pioneering work by Rosenblatt and colleagues established that the barrier function of the epithelial cell sheet is preserved even in the face of a mass cell death event triggered by acute UV irradiation (*Rosenblatt et al., 2001*). This observation suggests that the epithelium is capable of removing apoptotic cells while maintaining the continuity of TJs uninterrupted. However, the complex logistics of simultaneously breaking down the TJ between apoptotic and neighbor cells and assembling them anew between formally non-interacting cells are not sufficiently known. We first examined claudin dynamics among relevant cells during apoptotic cell elimination in order to understand how these incongruous processes are carried out. Apoptosis was induced in a confluent monolayer of a representative epithelial cell line, EpH4 stably expressing GFP-claudin-3 by pulsed laser irradiation of the nucleus in a targeted cell. Apical extrusion of the apoptotic cell and bicellular accumulation of claudin between newly formed contacts were typically observed over the course of an hour (*Figure 1A and B* and *Video 1*). Throughout, GFP-claudin-3 remained enriched to some degree between the apoptotic and neighbor cells (*Figure 1B*, purple arrowheads). Additionally, linear accumulation of GFP-claudin-3 could be observed more basally by about 40 min post-irradiation and steadily grew more pronounced over time (*Figure 1B*, green arrowheads). Immunofluorescence staining showed that other TJ proteins, such as ZO-1 and occludin, colocalize with the basally-enriched claudin, indicating the de novo formation of functional TJs between the newly neighboring cells (*Figure 1C*, green arrowheads) before the complete loss of the apoptotic cell-neighbor cell TJs (*Figure 1C*, purple arrowheads). Together, these data demonstrate that old and new TJs exist concurrently among the cells involved in apoptotic cell removal to ensure that there is no period of TJ absence that could compromise the epithelial barrier.

Previous studies have established that RhoA is activated in the neighbor cells at the interface with the apoptotic cell to enhance the contractility of the actomyosin purse string that extrudes the apoptotic cell (*Duszyc et al., 2021*). RhoA is also required for the maturation of epithelial cell adhesions through its effect on the underlying circumferential actomyosin and recent reports implicate it in the repair of TJ microinjuries (*Stephenson et al., 2019*). We, therefore, hypothesized that RhoA could play a role in the rapid de novo TJ formation, such as we see during apoptotic cell removal. Observation of active RhoA in neighbor cells using an effector domain biosensor (GFP-AnillinC) showed sustained activation at the apoptotic cell-neighbor cell interface as previously reported (*Figure 1D* and *Video 2*, white arrowheads). Importantly, RhoA activity persisted at newly formed cell contacts after full expulsion of the apoptotic cell (*Figure 1D*, yellow arrowheads). Furthermore, immunostaining revealed phosphorylation of the myosin regulatory light chain (ppMLC), a substrate of the RhoA effector ROCK, at such contacts that are clearly located beneath the extruded cell (*Figure 1E*). These data collectively point to the presence of an active pool of RhoA at the new TJs.

Apoptotic cells rapidly contract, causing the neighbor cells to extend lamellipodia, which acts to seal the basal surface. Since contact between protrusions from separated cells is necessary to initiate adhesion formation, we wondered if these lamellipodia alone, previously shown to be independent of both the Rho-ROCK pathway and myosin contractility, could instigate claudin enrichment at newly formed contacts (*Bonfim-Melo et al., 2022*). By imaging the plasma membrane (mScarlet-PLCδ PH) together with GFP-claudin-3 under ROCK inhibition, we found that claudin no longer accumulated at bicellular contacts once the old apoptotic cell-neighbor cell TJs broke down, despite the completion of basal closure by neighbor cell lamellipodia (*Figure 1F* and *Video 3*). This observation suggests that de novo TJ formation is impaired by ROCK inhibition. Accordingly, inducing mass apoptosis in the

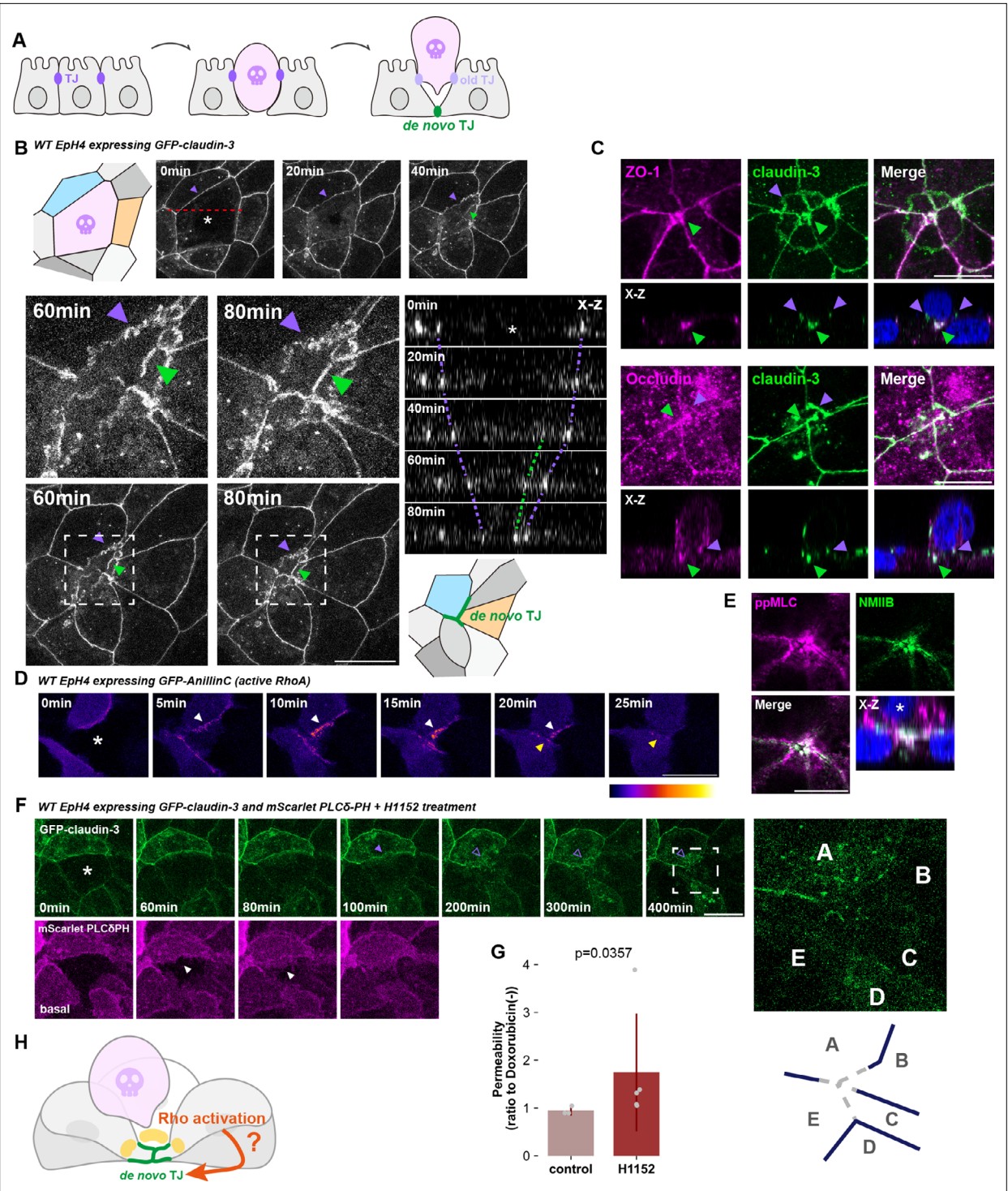

**Figure 1.** Rho-ROCK controls de novo tight junction assembly between newly adjacent cells during apical extrusion. (**A**) Schematic showing the elimination of an apoptotic cell by apical extrusion and de novo tight junction (TJ) formation. Tight junctions are established between newly adjacent cells (shown in green) concurrent with extrusion of the apoptotic cell. The TJs between the apoptotic cell and the cells that affect extrusion (shown in purple) gradually diminish as the new TJs mature, ensuring that the epithelial barrier is never disrupted. (**B**) Live cell images of EpH4 cells expressing GFP-claudin-3 in cells neighboring laser-wounded cell. Cell marked by an asterisk was wounded at time zero. The schematics represent the cells before laser irradiation and after apical extrusion. Purple arrowheads indicate the old TJ with the apoptotic cell and green arrowheads indicate the newly assembled TJ. The upper images (time = 60 and 80 min) show higher magnification corresponding to each time point. The right panels are orthogonal views. Scale bar = 20 μm. (See also **Video 1**). (**C**) Immunofluorescence images showing apical extrusion at 60 min post-laser irradiation. Cells were stained with anti-claudin-3 pAb (green) and either anti-ZO-1 mAb (magenta, upper) or anti-occludin mAb (magenta, lower). Purple arrowheads indicate

*Figure 1 continued on next page*

*Figure 1 continued*

old TJs with the apoptotic cell and green arrowheads indicate newly assembled TJs. The bottom panels represent orthogonal views. Scale bar = 10 μm. (**D**) Live cell images of cells surrounding a laser-wounded cell expressing GFP-AnillinC as a probe for active RhoA. Cell marked by an asterisk was wounded at 0 min. The arrowheads indicate accumulation of active RhoA at the junction between an apoptotic cell and a neighboring cell (white) or between newly adjacent cells (yellow). Scale bar = 20 μm. (See also *Video 2*). (**E**) Immunofluorescence images showing apical extrusion at 60 min post-laser irradiation. Cells were stained with phosphorylated myosin light chain (ppMLC) mAb (magenta) and anti-NMIIB mAb (green). The asterisk in the X-Z image indicates the apoptotic cell. Scale bar = 10 μm. (**F**) Live cell images of cells surrounding a laser-wounded cell co-expressing GFP-claudin-3 and mScarlet-PLCδPH (membrane marker). Cells were pre-treated with H1152 and the cell marked by an asterisk was wounded at 0 min. The filled purple arrowhead indicates a TJ with an apoptotic cell and the unfilled purple arrowheads indicate the gradual dissolution of said TJ. Filled purple arrowheads indicate TJ between an apoptotic cell and neighboring cells, while unfilled purple arrowheads indicate the disappearance of TJ. Images of scarlet-PLCδ PH are projections of basal confocal slices and the extending lamellipodia are indicated by the white arrowheads. The inset enlarged at right—and illustrated below—shows the absence of new TJs among the now adjacent cells A-E. Scale bar = 20 μm. (See also *Video 3*). (**G**) Bar graph showing the paracellular flux of 70 kDa FITC-dextran tracer molecule at 9 hr post-doxorubicin treatment (4 μM) (control: N=3, H1152: N=5; error bar: ± SD; Mann–Whitney U test). (**H**) Schematic illustrating the formation of new TJs between neighbor cells concurrent with extrusion of the apoptotic cell. The formation of these TJ requires the activation of RhoA.

epithelial monolayer by treatment with the anticancer drug doxorubicin caused a significant increase in the leakage of 70 kDa FITC-Dextran when ROCK was simultaneously inhibited, compared to the control condition (*Figure 1G*). Signaling through Rho-ROCK is clearly required for apical extrusion, since the apoptotic cell typically remains loosely embedded in the cell sheet. However, these results collectively hint that its importance in epithelial barrier homeostasis associated with apoptotic cell removal is not limited to this process: it is also critically important for claudin enrichment and/or engagement at new cell-cell contacts, the failure of which precludes the de novo formation of TJs (*Figure 1H*).

## De novo TJ formation in epidermis requires activation of the Rho-ROCK pathway

Given the technical difficulty in independently assessing the contribution of the Rho-ROCK pathway in the two intertwined events arising in cells neighboring an apoptotic cell—increased contractility to effect apical extrusion and de novo TJ formation to maintain the epithelial barrier—we chose to investigate its involvement in the context of de novo TJ formation in keratinocyte differentiation. The human keratinocyte cell line HaCaT differentiates into granular-like cells capable of forming TJs when cultured for 24 hr in a medium supplemented with a high concentration of calcium ion (9.8 mM) and a JNK inhibitor (Ca +JNK inh medium) (*Figure 2A*; *Aono and Hirai, 2008*; *Kitagawa et al., 2014*). In undifferentiated HaCaT cells, both claudin-1 and the AJ receptor E-cadherin were broadly distributed throughout the plasma membrane and their respective scaffolding proteins ZO-1 and α-catenin appeared cytoplasmic, indicating the absence of functional epithelial adhesions (*Figure 2B*, control). Following differentiation, claudin-1 and ZO-1 were sharply enriched at apical intercellular adhesion sites and the AJ components co-localized at the basolateral membrane (*Figure 2B*, Ca +JNKinh).

Accordingly, in differentiated HaCaT cells, the actin cytoskeleton was reorganized to form the circumferential actin ring associated with apical junctions (*Figure 2B*, F-actin). The morphological

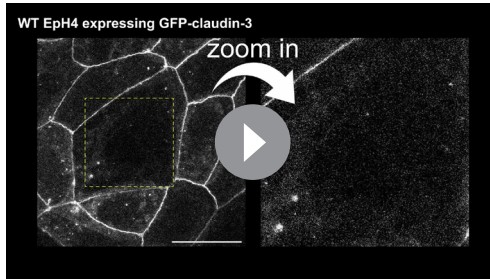

**Video 1.** EpH4 cells expressing GFP-claudin-3 in cells neighboring laser-wounded cells were imaged. Frames were taken every 1 min. (Scale bar, 20 μm.)

https://elifesciences.org/articles/102794/figures#video1

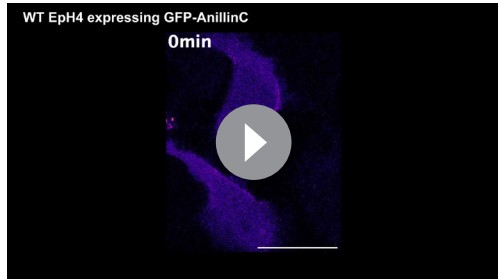

**Video 2.** EpH4 cells expressing GFP-AnillinC in cells neighboring laser-wounded cells were imaged. Frames were taken every 1 min. (Scale bar, 20 μm.)

https://elifesciences.org/articles/102794/figures#video2

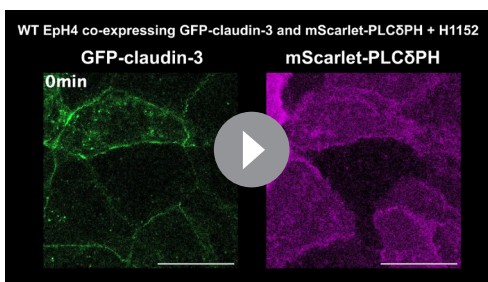

**Video 3.** EpH4 cells co-expressing GFP-claudin-3 and mScarlet-PLCδPH (membrane marker) in cells neighboring laser-wounded were imaged. Cells were pre-treated with H1152 (ROCK inhibitor). Frames were taken every 10 min. (Scale bar, 20 μm.)
https://elifesciences.org/articles/102794/figures#video3

changes observed here align well with those seen in differentiated keratinocytes of the SG2 layer in vivo. Of note, western blotting revealed that there was no significant difference in claudin-1 expression level before and after differentiation (*Figure 2C*). Moreover, in the mouse ear epidermis, claudin-1 was present across the entire plasma membrane in the spinous layers before differentiation into the SG2 layer (*Figure 2D*). Altogether, these observations suggest the existence of a regulatory mechanism that specifically instigates de novo TJ formation in keratinocytes of the SG2 layer, independent of transcriptional and/or translational control.

We then hypothesized the involvement of the Rho-ROCK pathway in de novo TJ formation associated with keratinocyte differentiation, similar to its role in de novo TJ formation during apoptotic cell elimination in monolayer epithelial cells. First, we investigated whether RhoA is activated during keratinocyte differentiation. We detected a significant increase in RhoA activation upon differentiation (*Figure 3A and B*). In differentiated HaCaT cells, phosphorylated MLCs were prominently observed at cell-cell contacts (*Figure 3C and D*). Similarly, in the mouse epidermis, ppMLC was notably enhanced only at apical junctions of keratinocytes in the SG2 layer (*Figure 3E*). These findings indicate an augmented Rho-ROCK pathway in keratinocytes of the granular layer during differentiation, consistent with a previous report that junctional tension is necessary for TJ formation in the granular layer (*Rübsam et al., 2017*).

We next examined whether activation of the Rho-ROCK pathway during differentiation is necessary for the formation of de novo TJs upon keratinocyte differentiation. When ROCK activity was inhibited by treatment with Y27632, claudin remained uniformly distributed across the entire plasma membrane, indicating impairment of TJ formation (*Figure 3F and G*). In contrast, E-cadherin signals at apical junctions were not affected by ROCK inhibition in differentiated HaCaT cells (*Figure 3F*). As we saw in monolayer epithelia, these results suggest that Rho-ROCK signaling specifically regulates de novo TJ formation in differentiating keratinocytes of the stratified epithelia by enabling claudin enrichment at intercellular adhesions, i.e., shifting claudin from an unpolymerized state in the basolateral membrane to a polymerized state in TJ strands (*Figure 3H*).

## Activation of ROCK alone can induce TJ formation in undifferentiated keratinocytes

Then, how does the Rho-ROCK pathway regulate the polymerization state of claudin? Since HaCaT cells already abundantly express claudins prior to differentiation, we wondered if activation of RhoA or ROCK alone could induce de novo TJ formation in the absence of differentiation factors. Overexpression of a constitutively active mutant of RhoA (Q61L, RhoA CA) led to the accumulation of claudin-3 at cell-cell contacts but only when both cells expressed RhoA CA (*Figure 4A*). By contrast, claudin-3 enrichment was not observed in cells expressing the other main Rho family GTPases, Rac1 and Cdc42 (*Figure 4—figure supplement 1A*). Importantly, additional TJ components, such as ZO-1 and the transmembrane protein occludin, were recruited to claudin-enriched cell-cell contacts and actin reorganization was also observed, indicating that these structures were highly TJ-like (*Figure 4A and B*). Moreover, activation of endogenous RhoA by treatment with the bacterial toxin cytotoxic necrotizing factor (Rho activator; *Figure 4C and G*), overexpression of a constitutively active mutant of ROCK (ROCKdelC; *Figure 4—figure supplement 1B*) and treatment with the ROCK activator Narciclasine also induced strong accumulation of claudin-3 in undifferentiated HaCaT cells (*Figure 4E and G*) to further support the proposition that signaling through RhoA-ROCK enables claudin polymerization. In some instances, the span of enriched claudin was significantly expanded laterally, indicating that the normally unpolymerized pool of claudin was being polymerized in situ, in the absence of positional cues (*Figure 4D and F*).

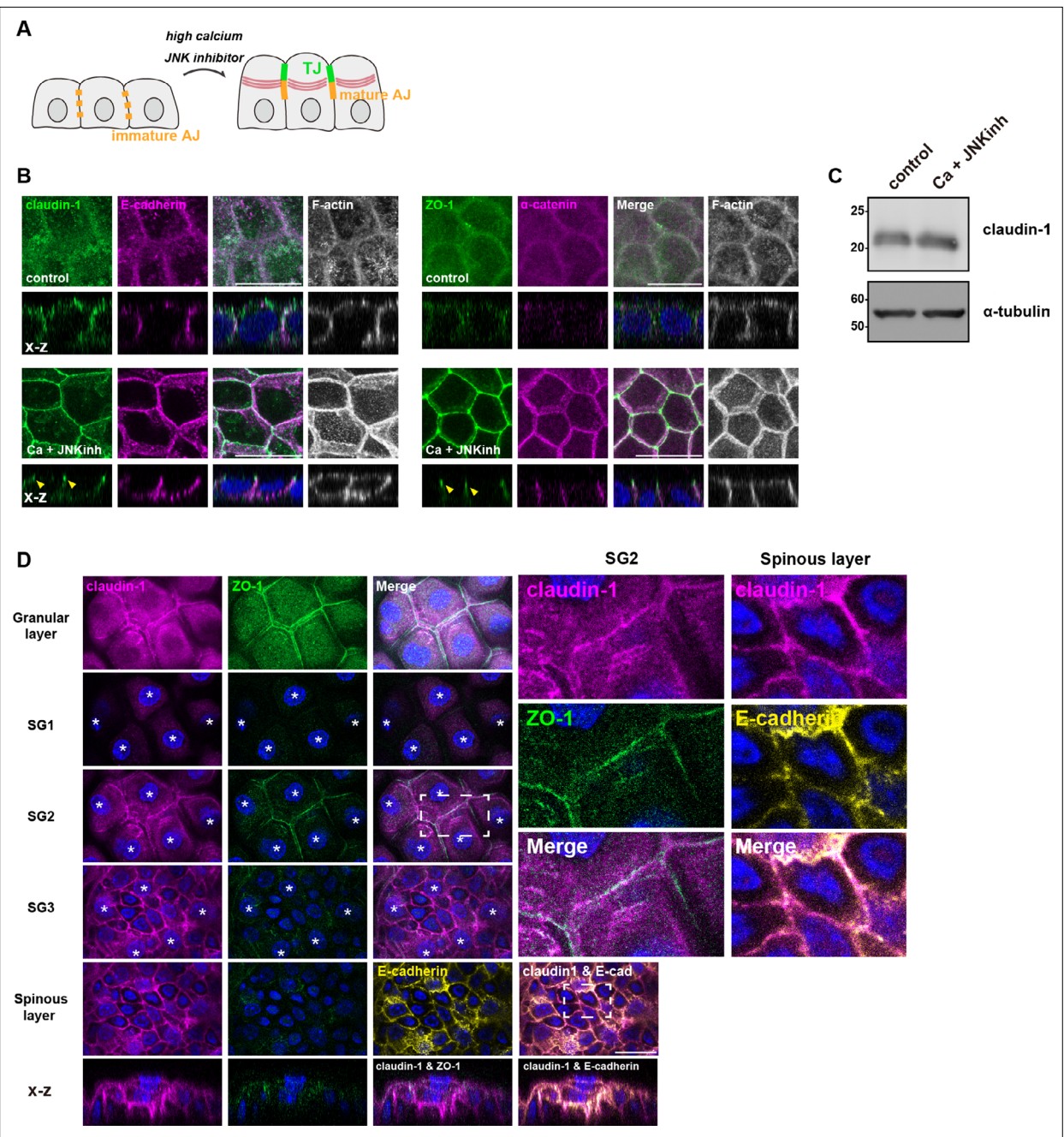

**Figure 2.** Claudins in epidermal cells are present at the plasma membrane in an unpolymerized state before differentiation in the granular layer. (**A**) Schematic showing the differentiation of human keratinocyte (HaCaT cells) into granular-like cells in high-calcium (9.8 mM) medium supplemented with a JNK inhibitor (40 μM) (Ca +JNK inh medium) for 24 hr. (**B**) HaCaT cells were cultured in normal medium (control) or Ca +JNK inh medium for 24 hr, fixed, and then stained with anti-claudin-1 pAb (green), anti-E-cadherin mAb (magenta) and phalloidin (grayscale; left) or anti-ZO-1 mAb (green), anti-alpha-catenin (magenta), and phalloidin (grayscale; right). X-Z images are shown below. Yellow arrowheads indicate tight junctions (TJs). Scale bar = 20 μm. (**C**) Whole-cell lysates of cells cultured in normal medium or Ca +JNK inh medium for 24 hr were immunoblotted with the indicated antibodies. Molecular weight measurements are in kDa. (**D**) Mouse ear whole-mount immunofluorescence analysis for claudin-1, ZO-1, and E-cadherin. The asterisk indicates the nucleus of cells belonging to that layer. The panels in the bottom row represent orthogonal views. Scale bar = 20 μm.

The online version of this article includes the following source data for figure 2:

**Source data 1.** Full blot original data for the Western blot shown in *Figure 2C*.

**Source data 2.** Full blot data for the Western blot shown in *Figure 2C*.

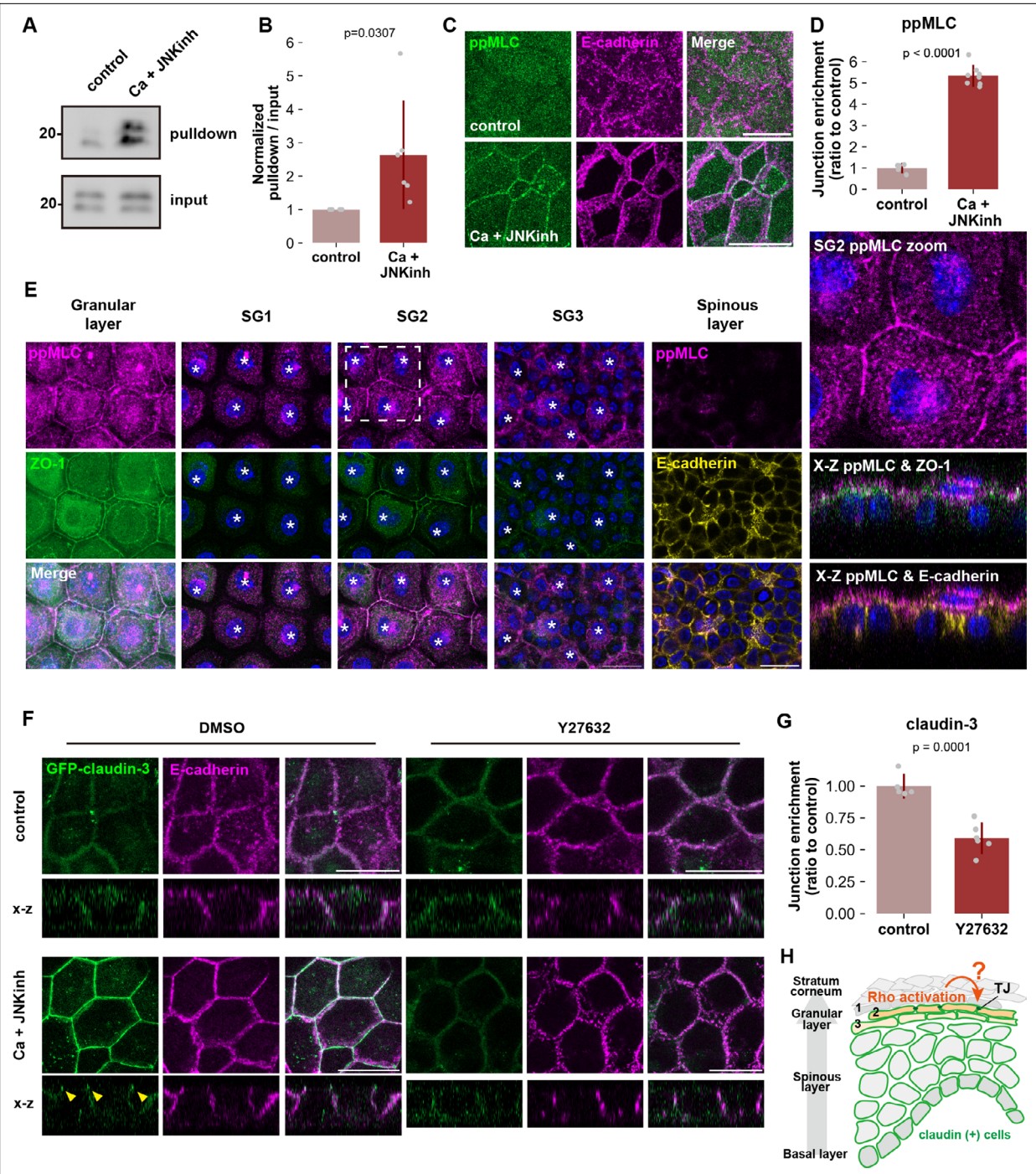

**Figure 3.** De novo tight junction (TJ) formation in the SG2 layer requires activation of Rho-ROCK. (**A** and **B**) RhoA activities in undifferentiated and differentiated HaCaT cells. Cells were lysed and subjected to a GST-Rhotekin pulldown assay. Total cell lysates and precipitates were analyzed by immunoblotting with anti-RhoA mAb. Molecular weight measurements are in kDa. (**B**) is a graph illustrating the rate of RhoA activities. RhoA activity was calculated by dividing the intensity of activated RhoA (pulldown signal) by the intensity of input signal (N=6; error bar: ± SD; Student's t-test). (**C** and **D**) HaCaT cells were cultured in normal medium or Ca +JNK inh medium for 24 hr, fixed, and then stained with anti-ppMLC mAb (green) and anti–E-cadherin mAb (magenta). Scale bar: 20 μm. Graph showing the quantification of junctional enrichment of phosphorylated myosin light chain (ppMLC) signals (**D**). The quantification methods are illustrated in *Figure 3—figure supplement 1* and details are described in Methods (control: N=5, Ca +JNK inh: N=8; error bar: ± SD; Student's t test). (**E**) Mouse ear whole-mount immunofluorescence analysis for ppMLC, ZO-1, and E-cadherin. The asterisk indicates the nucleus of cells belonging to the indicated layer. (**F** and **G**) HaCaT cells expressing GFP-claudin-3 were cultured in normal medium or Ca +JNK inh medium supplemented with DMSO (control) or Y27632 (ROCK inhibitor) for 24 hr, fixed, and then stained with anti–E-cadherin mAb (magenta). Yellow arrowheads indicate TJs. Scale bar: 20 μm. (**G**) is a bar graph illustrating the junctional enrichment of claudin-3 after differentiation

*Figure 3 continued on next page*

*Figure 3 continued*

(control: N=5, Ca +JNK inh: N=6; error bar:± SD; Student's t-test). (**H**) Schematic illustrating TJ formation in the epidermal granular layer. In the epidermis, all keratinocytes of basal, spinous, and granular layers express claudin-1. However, functional TJs are only formed in the SG2 layer. The activation of the Rho-ROCK pathway is crucial for the formation of TJs in the SG2 layer.

The online version of this article includes the following source data and figure supplement(s) for figure 3:

**Source data 1.** Full blot original data for the Western blot shown in *Figure 3A*.

**Source data 2.** Full blot data for the Western blot shown in *Figure 3A*.

**Figure supplement 1.** Details of the image analysis for quantification of junctional enrichment.

We previously showed that polymerized claudin is preferentially partitioned into detergent-resistant membranes (DRMs) enriched in cholesterol and very long-chain fatty acid-containing sphingomyelin (*Shigetomi et al., 2018*). Since both Rho activator and Narciclasine treatments can efficiently enrich claudin at cell-cell contacts, we next evaluated the polymerization state of claudin in treated cells biochemically. Rho activator and Narciclasine treatments markedly increased the amount of claudin-1 partitioned into the DRM fraction compared to the control condition. In contrast, in undifferentiated HaCaT cells, most of claudin-1 was solubilized by detergents, reflecting that claudin is present in an unpolymerized state in undifferentiated HaCaT cells (*Figure 4H and I*). These findings altogether indicate that activation of the Rho-ROCK pathway induces the transition of claudin from an unpolymerized state to a polymerized state.

## Matriptase activation by Rho-ROCK produces a pool of polymerizable claudin via TROP2 cleavage

Having established that signaling through Rho-ROCK triggers claudin polymerization, we next asked what the mechanism is. The single transmembrane proteins EpCAM and TROP2 bind to claudin and are abundantly expressed in epithelial cells and keratinocytes. They are thought to protect unpolymerized claudin from degradation to maintain a stable reserve apart from the TJ strand (*Wu et al., 2013*). This prompted us to consider whether they could play a greater role, not only in protecting unpolymerized claudin from degradation but also in inhibiting unregulated claudin polymerization by interfering with the interaction between claudins. Matriptase, a transmembrane serine protease, cleaves the extracellular region of EpCAM and TROP2 to sever their interaction with claudin (*Wu et al., 2017*; *Wu et al., 2020*). It was recently reported that, of the two, TROP2 plays a greater role in the epidermis (*Wu et al., 2020*). Therefore, we reasoned that Rho-ROCK signaling releases TROP2-bound claudin by activating matriptase.

Matriptase is normally present at the plasma membrane in an inactive state but it is activated by autocleavage in response to external signals. After cleavage of the substrate, the Kunitz-type protease inhibitor family proteins Hai1 and Hai2 immediately bind to matriptase to block its activity (*Friis et al., 2014*; *Chiu et al., 2022*; *Larsen et al., 2013*). The extracellular region of the matriptase-Hai complex is then cleaved and released from the cell (*Figure 5A*, illustration) (*List et al., 2006*). To assess the activation state of Matriptase, we employed two mouse monoclonal antibodies: M24, which detects total Matriptase protein levels, and M69, which specifically recognizes the activated form of matriptase. These antibodies have been widely used in previous studies on matriptase (*Benaud et al., 2001*; *Benaud et al., 2002*; *Chen et al., 2010*; *Gaymon et al., 2023*; *Tseng et al., 2010*). One such study showed that matriptase is activated by simply placing cells in an acidic medium at pH 6.0 for 20 min (*Tseng et al., 2010*). Addition of acid medium to undifferentiated HaCaT cells decreased the total amount of matriptase and increased the amount of cleaved TROP2 compared to the control, indicating that matriptase is activated and cleaves TROP2 (*Figure 5A*). This was accompanied by claudin accumulation at cell-cell contacts and an increase in insoluble, i.e., polymerized, claudin (*Figure 5C–F*). In an intriguing contrast to the results of Rho-ROCK activation by pharmacological means (*Figure 4C and E*) where we saw development of junction-associated F-actin and AJ maturation simultaneous with claudin polymerization, neither was observed here (*Figure 5C*). This observation suggests that bypassing Rho-ROCK and activating matriptase directly is sufficient to induce claudin strand assembly. However, treatment with Rho-ROCK activators increased matriptase activity and enhanced TROP2 cleavage, indicating that matriptase is a bona fide effector of Rho-ROCK signaling (*Figure 5A and B*). Furthermore, inhibition of matriptase activity with Camostat treatment suppressed claudin

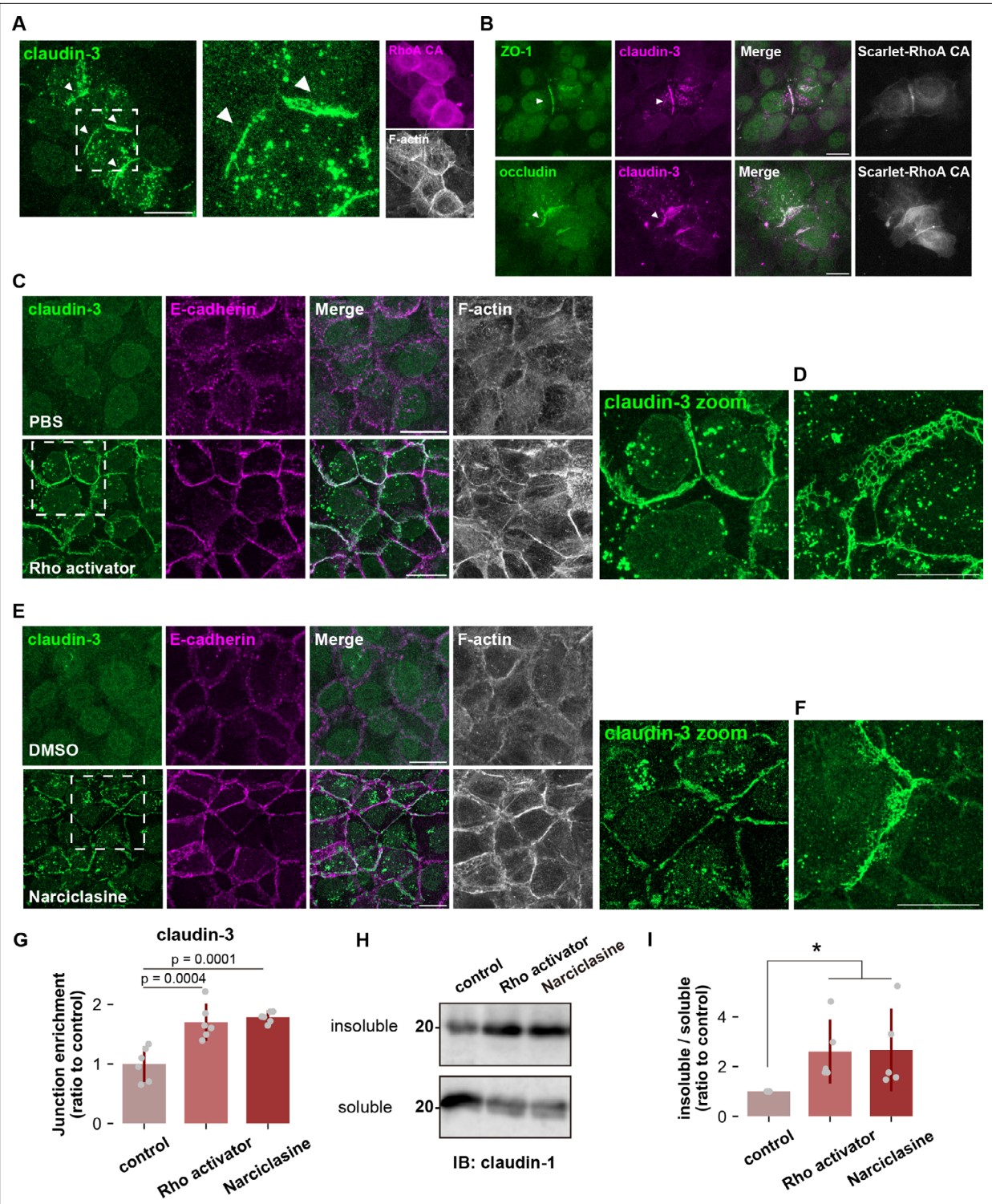

**Figure 4.** Rho-ROCK activation induces ectopic tight junction (TJ) formation in undifferentiated keratinocytes. (**A**) Undifferentiated HaCaT cells expressing constitutive active RhoA (RhoA CA) were stained with anti-claudin-3 pAb (green), and phalloidin (grayscale). The arrowheads indicate ectopic tight junctions. Scale bar: 20 μm. (**B**) Undifferentiated HaCaT cells expressing RhoA CA were stained with anti-claudin-3 pAb (magenta) and either anti-ZO-1 mAb (green; upper) or anti-occludin mAb (green; lower). The arrowheads indicate ectopic TJs. Scale bar: 20 μm. (**C** and **D**) Confluent undifferentiated HaCaT cells were cultured in serum-free medium for 24 hr, treated with a recombinant bacterial cytotoxic necrotizing factor (CNF) toxin (CN-03), also known as Rho activator II (1 μg/ml) for 2 h, fixed, and then stained with anti-claudin-3 pAb (green), anti-E-cadherin mAb (magenta), and phalloidin (grayscale). Scale bar: 20 μm. (**D**) shows an alternate field of view corresponding to the experiment in (**C**). (**E** and **F**) Confluent undifferentiated

*Figure 4 continued on next page*

*Figure 4 continued*

HaCaT cells were treated with DMSO (control) or Narciclasine (100 nM; ROCK activator) for 4 hr, fixed, and then stained with anti-claudin-3 pAb (green), anti-E-cadherin mAb (magenta), and phalloidin (grayscale). Scale bar: 20 µm. (**F**) shows an alternate field of view of the experiment in (**E**). (**G**) Junctional enrichment of claudin-3 was quantified based on the method described in *Figure 3—figure supplement 1* (N=6; error bar: ± SD; Tukey-Kramer One-way Anova). (**H** and **I**) Western blot of Triton X-100–soluble fractions and insoluble fractions from HaCaT cells treated with Rho activator (CN-03) or Narciclasine. The proportion of insoluble claudin-1 was quantified in (**I**) (N=5; error bar: ± SD; Kruskal-Wallis test followed by Steel-Dwass post hoc test; *p<0.05).

The online version of this article includes the following source data and figure supplement(s) for figure 4:

**Source data 1.** Full blot original image for the Western blot shown in *Figure 4H*.

**Source data 2.** Full blot data for the Western blot shown in *Figure 4H*.

**Figure supplement 1.** Claudin accumulation is induced by constitutive active ROCK but not by either constitutive active Rac or Cdc42.

accumulation at cell-cell contacts induced by Rho-ROCK activation, which further supports the notion that activation of matriptase occurs downstream of Rho-ROCK pathway (*Figure 5G and H*). Collectively, these results demonstrate that activation of matriptase by Rho-ROCK signaling liberates claudin from sequestration by TROP2 and acts as the rate limiting process in de novo TJ formation; indeed, TROP2 was uniformly present throughout the plasma membrane in undifferentiated cells but it was specifically excluded from the region where claudin was polymerized between cells expressing Rho CA cells (*Figure 5I*).

It was previously reported that the amount of claudin-1 protein is significantly reduced TROP2-depleted keratinocytes, which we confirmed (*Figure 6A and B*; *Wu et al., 2020*). More interestingly, however, quantification of claudin-1 surface expression revealed that the amount was almost halved in TROP2 KO cells, even accounting for decreased overall expression (*Figure 6C and D*). Moreover, claudin-1 was most prominently found at the Golgi apparatus of TROP2 KO cells, where it was distributed throughout the plasma membrane in undifferentiated wild-type cells, only appearing intermittently there as strongly enriched dots (*Figure 6E*). These punctate claudin-1 accumulations contained ZO-1 and occludin, indicating that claudin-1 is incorporated in a quasi-TJ state (*Figure 6—figure supplement 1*). In other words, claudin-1 cannot stably exist in an unpolymerized state at the plasma membrane in the absence of TROP2. As expected by the lack of available surface claudin, neither Rho-ROCK activation nor matriptase activation by acid medium treatment could induce claudin polymerization in TROP2 KO cells (*Figure 6F–H*). Taken together, these findings indicate that the regulation of claudin by TROP2 is not limited to stabilizing unpolymerized claudin at the plasma membrane: it begins at the Golgi apparatus where—acting as a chaperone—it mediates proper sorting and trafficking (*Figure 6I*).

The results so far strongly suggest that the Rho-ROCK pathway enables claudin polymerization, the initial step of de novo TJ formation, by activating matriptase, which frees the claudin pool sequestered by TROP2 at the lateral membrane; this is independent of its long-established effect on junctional maturation through actomyosin. We next performed a series of experiments to verify this hypothesis. First, we found that matriptase was rapidly activated in HaCaT cells introduced into the Ca +JNKinh medium to induce differentiation; degradation of TROP2 was observed in parallel (*Figure 7A and B*). Immunostaining with the M24 antibody that detects total protein revealed that matriptase is constitutively distributed along the lateral membrane, below TJs, regardless of HaCaT cell differentiation state (*Figure 7C*). By contrast, staining with the M69 antibody, which specifically recognizes the activated form, revealed no signal in undifferentiated cells but following differentiation, activated matriptase could be widely detected at the lateral membrane (*Figure 7D and E*). These observations indicate that while the subcellular localization of matriptase remains unchanged during differentiation, it undergoes activation at the lateral membrane in differentiated granular-layer-like cells.

We then examined HaCaT differentiation in the presence of the matriptase inhibitor Camostat. Consistent with the result of our experiment with Rho CA overexpression, claudin failed to accumulate at cell-cell junctions in Camostat-treated cells (*Figure 7F and H*). A previous study demonstrated that an increase in intercellular tension at AJs is indispensable for the formation of TJs in the SG2 layer (*Rübsam et al., 2017*). To assess AJ tension, we stained with the α–18 monoclonal antibody, sensitive to the tension-dependent conformational change of α-catenin, a core AJ component. In control cells, differentiation into granular-layer-like cells led to a marked increase in α–18 signal at cell–cell adhesions. Importantly, when HaCaT cells were treated with Camostat to inhibit matriptase before

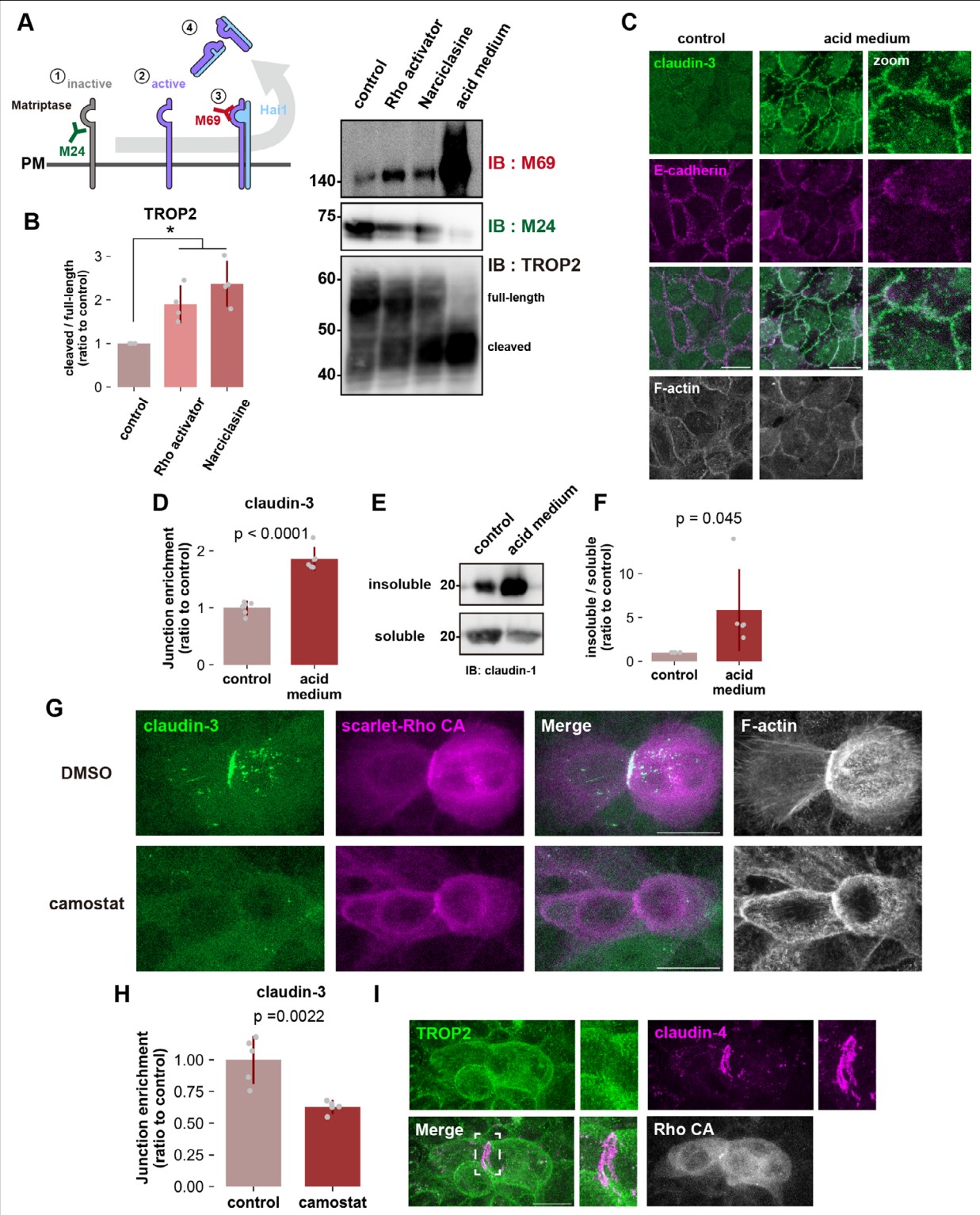

**Figure 5.** Rho-ROCK pathway acts upstream of matriptase activation in de novo tight junction (TJ) formation. (**A** and **B**) Whole-cell lysates of wild-type (WT) treated with Rho activator (CN-03), Narciclasine, or acid medium (pH 6.0) were immunoblotted with the indicated antibodies. Molecular weight measurements are in kDa. Schematic showing the sequential activation of matriptase. M24 mAb detects total Matriptase and M69 mAb the activated form. (**B**) Quantification of the cleaved TROP2 fragment relative to total TROP2. (N=4; error bar: ± SD; Kruskal-Wallis test followed by Steel-Dwass post hoc test; *p<0.05). (**C** and **D**) Confluent undifferentiated HaCaT cells were cultured in normal (control) or acid medium (pH 6.0) for 20 min, fixed, and then stained with anti-claudin-3 pAb (green), anti-E-cadherin mAb (magenta), and phalloidin (grayscale). Scale bar = 20 µm. (**D**) is a bar graph illustrating

*Figure 5 continued on next page*

*Figure 5 continued*

the junctional enrichment of claudin-3 (N=6; error bar: ± SD; Student's t-test). (**E** and **F**) Western blot of Triton X-100–soluble and insoluble fractions from confluent undifferentiated HaCaT cells cultured in acid medium. The proportion of insoluble claudin-1 was quantified in (**F**) (N=5; error bar: ± SD; Student's t-test). (**G**) Undifferentiated HaCaT cells treated with DMSO (control) or Camostat (serine protease inhibitor) expressing constitutive active RhoA (Rho CA) were fixed and stained with anti-claudin-3 pAb (green) and phalloidin (grayscale). Scale bar: 20 μm. (**H**) Bar graph illustrating the extent to which the matriptase inhibitor Camostat negated the Narciclasine-induced claudin enrichment at cell-cell contacts. After inhibition of serine protease with Camostat, confluent undifferentiated HaCaT cells were treated with Narciclasine (100 nM; ROCK activator) for 4 hr and stained to quantify the junctional enrichment of claudin-3 (N=6; error bar: ± SD; Student's t-test). (**I**) Undifferentiated HaCaT cells expressing constitutive active RhoA (Rho CA) were stained with anti-TROP2 mAb (green) and claudin-4 mAb (magenta). Insets are enlarged images. Scale bar: 20 μm.

The online version of this article includes the following source data for figure 5:

**Source data 1.** Full blot original images for the Western blot shown in *Figure 5A and E*.

**Source data 2.** Full blot data for the Western blot shown in *Figure 5A and E*.

induction, we observed an equivalent increase in α–18 signal at AJs (*Figure 7F*). However, we did not detect claudin enrichment at cell-cell contacts under these conditions (*Figure 7F and H*). These results suggest that matriptase inhibition specifically disrupts TJ formation during granular-layer differentiation without impeding AJ maturation.

Similarly, in HaCaT cells overexpressing Hai1, an endogenous inhibitor of matriptase, TJ formation was severely suppressed (*Figure 7G and H*). Lastly, we sought to extend our findings to the granular layer differentiation of the epidermis. As in undifferentiated HaCaT cells, TROP2 was distributed throughout the plasma membrane of the spinous layer but was progressively lost within the granular layer—particularly at the SG2 layer—as TJs appeared and matured (*Figure 7I*). These results suggest that ROCK-dependent activation of matriptase and cleavage of TROP2 by matriptase are crucial steps in the formation of de novo TJs associated with granule cell differentiation in the epidermis (*Figure 7J*).

## Rho-ROCK liberates EpCAM-sequestered claudin through matriptase to enable rapid de novo TJ formation during apoptotic cell elimination

We next investigated whether the same molecular mechanism of de novo TJ formation in stratified epithelium plays a role in the homeostasis of the epithelial barrier in monolayer epithelial cells.

It has been established that EpCAM, rather than TROP2, regulates claudin in simple epithelium (*Wu et al., 2013*; *Higashi et al., 2023*; *Lei et al., 2012*; *Wu et al., 2017*). Therefore, we established EpCAM KO EpH4 cells. In EpCAM KO cells, the amount of claudin-1 was significantly reduced, consistent with previous studies and TROP2 depletion in HaCaT cells (*Figure 8A and B*; *Wu et al., 2013*). Immunofluorescence with the standard Triton X-100 membrane permeabilization revealed a sharp, linear localization of claudin-4 at apical junctions in EpCAM KO cells that was indistinguishable from wild-type cells, giving the appearance that TJs are formed normally under steady-state conditions (*Figure 8C*). However, when the detergent was changed to digitonin, significant claudin-7 staining was observed in the lateral membranes in wild-type cells (*Figure 8D*). By contrast, the lateral pool of claudin-7 was severely reduced in EpCAM KO cells; as in HaCaT cells depleted of TROP2, claudin was seen to accumulate at the Golgi apparatus instead (*Figure 8D and E*). These results indicate that claudin transport is similarly regulated in simple epithelium and stratified epithelium by EpCAM and TROP2, respectively.

We then wondered whether cleavage of EpCAM by forcibly activating matriptase through treatment with acid medium would also induce de novo TJ formation here. When wild-type EpH4 cells cultured on permeable supports were treated with the acid medium on the basolateral side, claudin enrichment could be observed throughout the lateral membrane in contrast to EpCAM KO cells that showed no such ectopic TJ expansion (*Figure 8F*). These findings suggest that, similar to stratified epithelium, cleavage of EpCAM by matriptase supplies polymerizable claudins for de novo TJ formation in monolayer epithelium.

Finally, we investigated whether ROCK-dependent activation of matriptase and cleavage of EpCAM by matriptase is important for rapid de novo TJ formation during apoptotic cell elimination. A previous study showed that EpCAM cleavage mediates TJ repair at small leakages induced by acute tension elevation (*Higashi et al., 2023*). Apical extrusion of the apoptotic cell induced by pulsed laser irradiation proceeded without fail in EpCAM KO cells as in wild-type cells but the formation of de

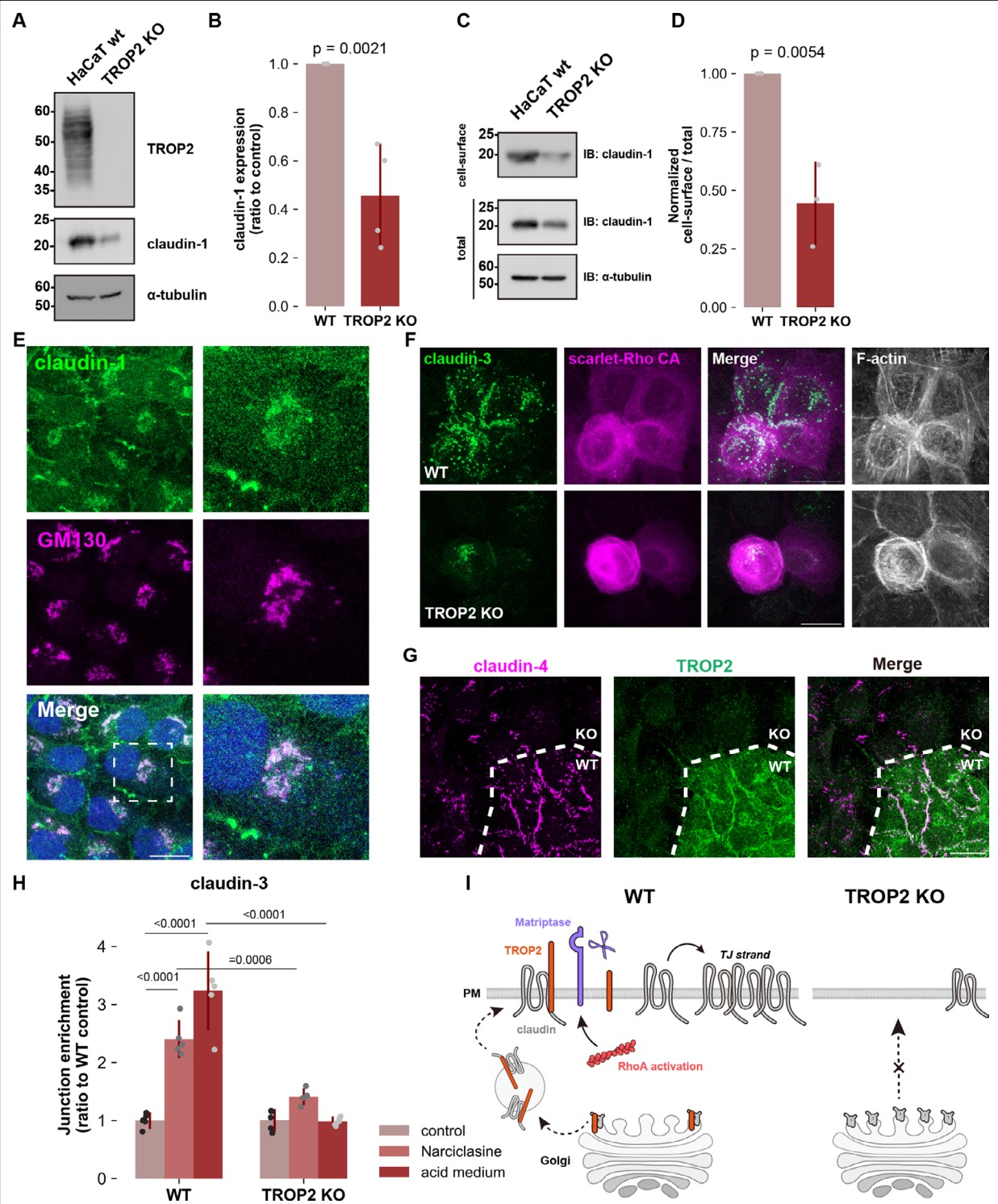

**Figure 6.** TROP2 is required for intracellular transport of claudin to the plasma membrane in keratinocytes. (**A** and **B**) Whole-cell lysates of wild-type (WT) and TROP2 KO HaCaT cells were immunoblotted with the indicated antibodies. Molecular weight measurements are in kDa. The expression level of claudin-1 normalized to α-tubulin was quantified in (**B**) (N=4; error bar: ± SD; Student's t test). (**C** and **D**) Representative western blots of surface and total amount of claudin-1. Molecular weight measurements are in kDa. The graph of (**D**) illustrates the quantification of claudin-1 present on the cell surface relative to the total amount (N=3; error bar: ± SD; Student's t test). (**E**) Undifferentiated TROP2 KO HaCaT cells were stained with anti-claudin-1 pAb (green) and anti-GM130 mAb (magenta; Golgi marker). Insets are enlarged images. Scale bar: 20 μm. (**F**) Undifferentiated WT or TROP2 KO HaCaT cells expressing constitutive active RhoA (Rho CA) were stained with anti-claudin-3 pAb (green) and phalloidin (grayscale). Scale bar:

*Figure 6 continued on next page*

*Figure 6 continued*

20 μm. (**G**) Undifferentiated WT and TROP2 KO HaCaT cells were co-cultured, incubated in acid buffer (pH 6.0) for 20 min, fixed, and then stained with anti-claudin-4 mAb (green) and anti-TROP2 mAb (magenta). Dotted line overlays the border between WT and TROP2 KO cells. Scale bar: 20 μm. (**H**) Bar graph illustrating the impairment of either Narciclasine- or acid medium-induced claudin-3 accumulation at cell-cell contacts by TROP2 depletion. Confluent undifferentiated WT or TROP2 KO HaCaT cells were treated with Narciclasine (ROCK activator) or acid medium (pH 6.0) and stained to quantify junctional enrichment of claudin-3 (N=5; error bar: ± SD; Tukey-Kramer One-way Anova). (**I**) Schematic illustrating the mechanism of ectopic TJ formation through RhoA activation. In wild-type keratinocytes, claudin and TROP2 form a complex in the Golgi apparatus and are subsequently transported to the plasma membrane. In contrast, in cells lacking TROP2, claudin, which cannot form a complex with TROP2, accumulates in the Golgi apparatus, leading to a significant decrease in the amount of claudin at the plasma membrane. In wild-type cells, TROP2 is cleaved by matriptase activated via the Rho-ROCK pathway. This cleavage results in the breakdown of the TROP2-claudin complex, allowing released claudins to form de novo tight junctions (TJs).

The online version of this article includes the following source data and figure supplement(s) for figure 6:

**Source data 1.** Full blot data for the Western blot shown in *Figure 6A and C*.

**Source data 2.** Full blot original images for the Western blot shown in *Figure 6A and C*.

**Figure supplement 1.** Other tight junction (TJ) components are co-localized with claudin puncta on the plasma membrane of undifferentiated TROP2 KO HaCaT cells.

novo TJs between newly adjacent cells was never observed (*Figure 8G* and *Video 4*). Furthermore, de novo TJ formation after apoptosis induction was likewise suppressed in EpH4 cells treated with Camostat or overexpressing Hai1 (*Figure 8H and I* and *Videos 5 and 6*). These findings support the notion that de novo TJ formation occurring in parallel with apical extrusion of apoptotic cells requires the degradation of EpCAM by matriptase.

The rapid restoration of TJ continuity upon the elimination of apoptotic cells is crucial for epithelial barrier homeostasis. Even when apoptosis was induced by treatment with 4 μM doxorubicin, leakage of 70 kDa FITC-Dextran was not detected in control EpH4 cells, indicating the preservation of epithelial barrier integrity. In clear contrast, significant leakage of 70 kDa FITC-Dextran was detected in EpCAM KO cells and epithelial cells in which matriptase function was inhibited (*Figure 8J*). Altogether, we conclude that Rho activation triggers matriptase-mediated EpCAM cleavage, which quickly introduces a supply of polymerizable claudin to enable the preservation of barrier integrity in damaged epithelial cell sheets.

## Discussion

In this paper, we demonstrate that the rapid formation of de novo TJs observed during the elimination of apoptotic cells in simple epithelium and the differentiation of granular cells in stratified epithelium begins with the activation of the Rho-ROCK pathway and follows a common series of reactions: (1) activation of matriptase, (2) destruction of the EpCAM/TROP2-claudin complex via cleavage of EpCAM/TROP2 by activated matriptase, and (3) liberated supply of polymerizable claudin (*Figure 9*).

The contribution of the Rho-ROCK pathway to the maintenance of epithelial barrier homeostasis has been explicated in various experimental contexts. It is known to increase the contractile force of cells adjacent to apoptotic cells to enable efficient apoptotic cell elimination (*Rosenblatt et al., 2001*; *Duszyc et al., 2021*; *Slattum et al., 2009*), and recent research using Xenopus embryos showed that local activation of Rho (referred to as Rho Flare) upregulates the contractility of TJ-associated actomyosin at the site of TJ rupture, minimizing TJ disruption and thus barrier breach (*Stephenson et al., 2019*). Here, we utilized monolayer and stratified epithelial cells, which enabled us to distinguish a novel role of signaling through Rho-ROCK, distinct from its effect on the cytoskeleton: to release the stockpiled claudin—complexed with EpCAM/TROP2—from the lateral membrane to promote rapid de novo TJ assembly. Several previous studies support our proposal. For one, overexpression of constitutively activated RhoA induces ectopic TJs at the lateral membrane (*Jou et al., 1998*; *Takaishi et al., 1997*). For a more physiologically relevant example, hyperosmotic stress induces rapid expansion of TJs at the lateral membrane (*Madara, 1983*; *Shiomi et al., 2015*). Considering that Rho-ROCK pathway is known to be hyper-activated by the hyperosmotic stress (*Thirone et al., 2009*), the mechanism of supplying polymerizable claudin via the Rho-ROCK pathway is likely to contribute to the adaptive expansion of TJs to hyperosmotic stress in epithelial cells.

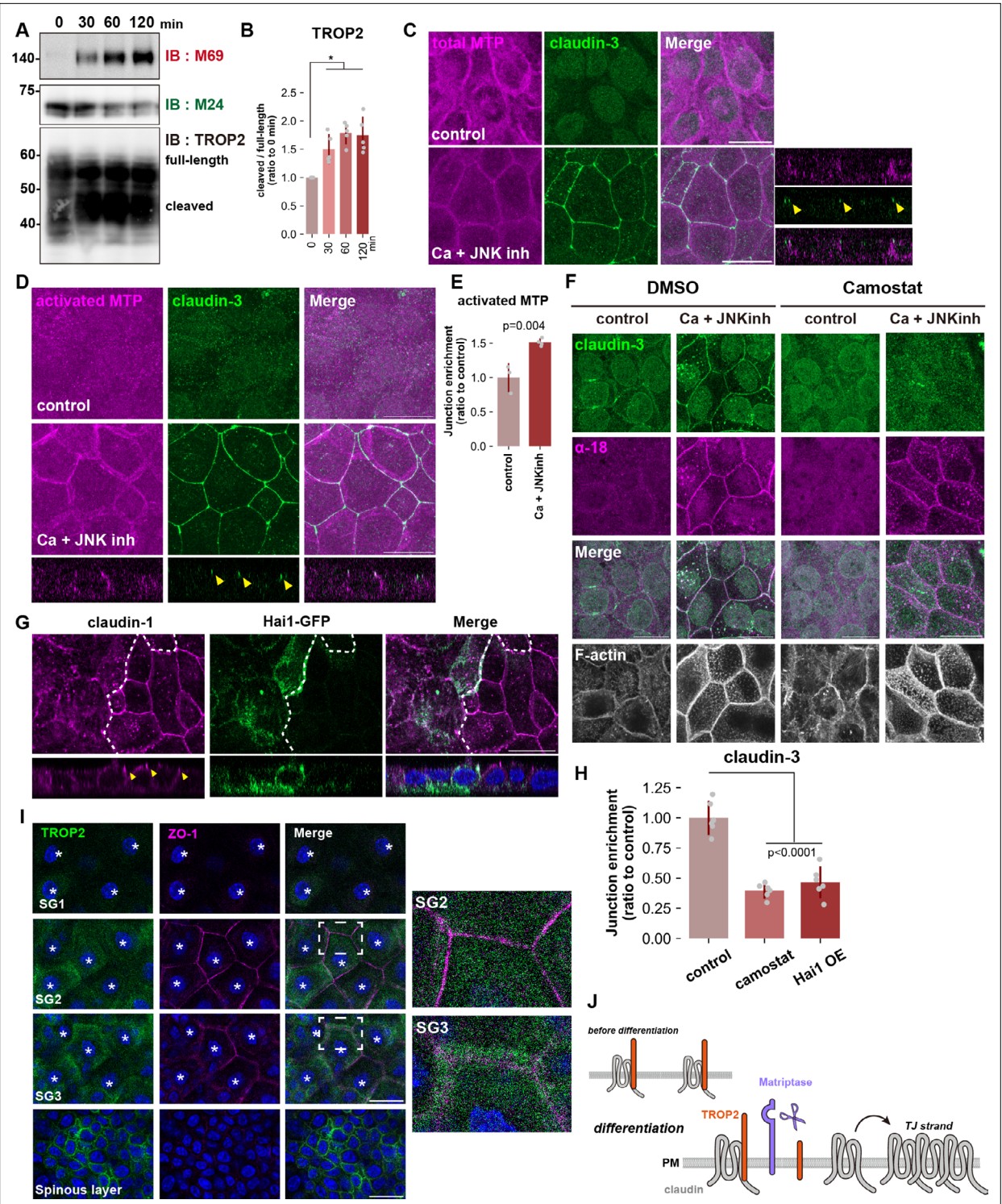

**Figure 7.** Matriptase cleaves TROP2 in the SG2 layer of epidermis. (**A**) Wild-type (WT) HaCaT cells were cultured in Ca +JNK inh medium for indicated time points and time-course change of indicated proteins was examined by western blotting. (N=5; error bar: ± SD; Kruskal-Wallis test followed by Steel-Dwass post hoc test; *p<0.05). (**B**) Quantification of the cleaved TROP2 fragment relative to total TROP2. (N=5; error bar: ± SD; Kruskal-Wallis test followed by Steel-Dwass post hoc test; *p<0.05). (**C**, **D**, and **E**) HaCaT cells were cultured in normal medium or Ca +JNK inh medium for 24 hr, fixed, and then stained with the m24 mAb (total matriptase: magenta) and anti-claudin-3 pAb (green; **C**) or with the M69 mAb (activated matriptase: magenta) and anti-claudin-3 pAb (green; **D**). Scale bar = 20 µm. (**E**) is a bar graph illustrating the junctional enrichment of activated matriptase (control: N=3, Ca +JNK inh: N=4; error bar: ± SD; Student's t-test). (**F**) HaCaT cells were cultured in normal medium or Ca +JNK inh medium supplemented with DMSO (control) or Camostat (serine protease inhibitor) for 24 hr, fixed, and then stained with anti-claudin-3 pAb (green), anti-α-catenin mAb (α–18;

*Figure 7 continued on next page*

*Figure 7 continued*

magenta). Scale bar: 20 µm. (**G**) WT HaCaT cells and cells over-expressing Hai1-GFP were co-cultured in Ca +JNK inh medium for 24 hr, fixed, and then stained with anti-claudin-1 pAb. Scale bar: 20 µm. (**H**) Bar graph illustrating the effects of either Camostat treatment or over-expression of Hai1 on accumulation of claudin-3 at cell-cell contacts in HaCaT cells cultured in Ca +JNK inh medium (N=6; error bar: ± SD; Tukey-Kramer One-way Anova). (**I**) Mouse ear whole-mount immunofluorescence analysis for TROP2 and ZO-1. The asterisk indicates the nucleus of cells belonging to the indicated layer. (**J**) Schematic illustrating the mechanism of TJ formation during differentiation into the granular layer in the epidermis. In undifferentiated cells, claudin is expressed and bound to TROP2 on the plasma membrane. As cells differentiate into the granular layer, RhoA becomes activated, followed by the activation of matriptase, leading to the cleavage of TROP2. The cleavage of TROP2 results in the dissociation of claudin, and the dissociated claudins then polymerize into de novo TJs.

The online version of this article includes the following source data for figure 7:

**Source data 1.** Full blot data for the Western blot shown in *Figure 7A*.

**Source data 2.** Full blot original image for the Western blot shown in *Figure 7A*.

While our results make clear that the cascade of reactions leading to claudin release is triggered by Rho-ROCK-dependent activation of matriptase, the precise mechanism of this step is unclear. A previous study showed that matriptase binds the actin cross-linking protein filamin, the depletion of which severely impairs its activation (*Kim et al., 2005*). Therefore, it is conceivable that actin reorganization associated with Rho-ROCK activation allows filamin to engage matriptase and activate it. Alternatively, Rho-ROCK may alter signaling pathways that regulate matriptase activity. Matriptase is intrinsically inhibited by Hai1; Hai1 is cleaved by the membrane type 1 matrix metalloproteases MMP14 and MMP7, which can stably activate matriptase (*Domoto et al., 2012*; *Ishikawa et al., 2017*). Notably, MMP14 is targeted to invadopodia in invasive cancer cells via RhoA-dependent exocytosis (*Sakurai-Yageta et al., 2008*). The Rho-ROCK pathway may similarly facilitate the delivery of MMP14 and/or MMP7 to the plasma membrane to activate matriptase through Hai1 cleavage. Still another possibility involves the type II transmembrane serine protease Tmprss2, the loss of which results in a significant decrease in matriptase expression and disruption of the TJ barrier but the relationship between Rho-ROCK and Tmprss2 is unknown (*Rickman et al., 2024*). Clearly, further studies are needed to elucidate how the Rho-ROCK pathway activates matriptase.

Our analysis focused on matriptase as the protease that cleaves EpCAM/TROP2, but other proteases that process EpCAM/TROP2 have been reported. Recently, Higashi et al. showed that membrane-anchored serine proteinases (MASPs) cleave EpCAM (*Higashi et al., 2023*); MASPs are present at the apical surface in polarized epithelial cells. Rupture of TJs results in a partial loss of the TJ fence function, leading to a change in the localization of MASPs from the apical membrane to the lateral membrane. This allows MASPs to cleave the EpCAM-claudin complex at the lateral membrane. By contrast, we found that matriptase is always present in the lateral membrane and actively regulates de novo TJ formation under the control of Rho-ROCK signaling. Whether these two pathways work independently or in parallel, or whether each of these pathway functions only in specific tissues or contexts, needs to be investigated in the future. Additionally, EpCAM is reportedly cleaved by A disintegrin and metalloprotease (ADAM) 10 and ADAM17 but it is unknown whether this contributes to de novo TJ formation (*Maetzel et al., 2009*). In the present study, overexpression of Hai1, which specifically inhibits matriptase activity, inhibited the formation of de novo TJs in both apoptotic cell elimination and granular layer differentiation, suggesting that the primary means of freeing sequestered claudin from EpCAM/TROP2 in these contexts is through matriptase.

In this paper, we show that Rho-ROCK-dependent matriptase activation enables de novo TJ formation in the SG2 layer of the epidermis. Prior studies have shown that applying the ROCK activator Narciclasine to mice with psoriatic dermatitis reduces inflammation (*Kong et al., 2022*). Topical application of oleoyl-l-α-lysophosphatidic acid (LPA), a potent Rho activator, was reported to increase the skin barrier in mice (*Sumitomo et al., 2019*). However, the molecular basis for the activation of Rho-ROCK pathways to improve the epidermal barrier was not elucidated. Our findings provide a rationale for the validity of these treatments.

Finally, we revealed that EpCAM/TROP2, besides its role in maintaining a stable pool of unpolymerized claudin at the plasma membrane, functions as a chaperone to facilitate intracellular transport of claudin from the Golgi apparatus to the plasma membrane. The molecular mechanism of claudin transport to the plasma membrane remains largely unknown. In the future, it will be necessary to

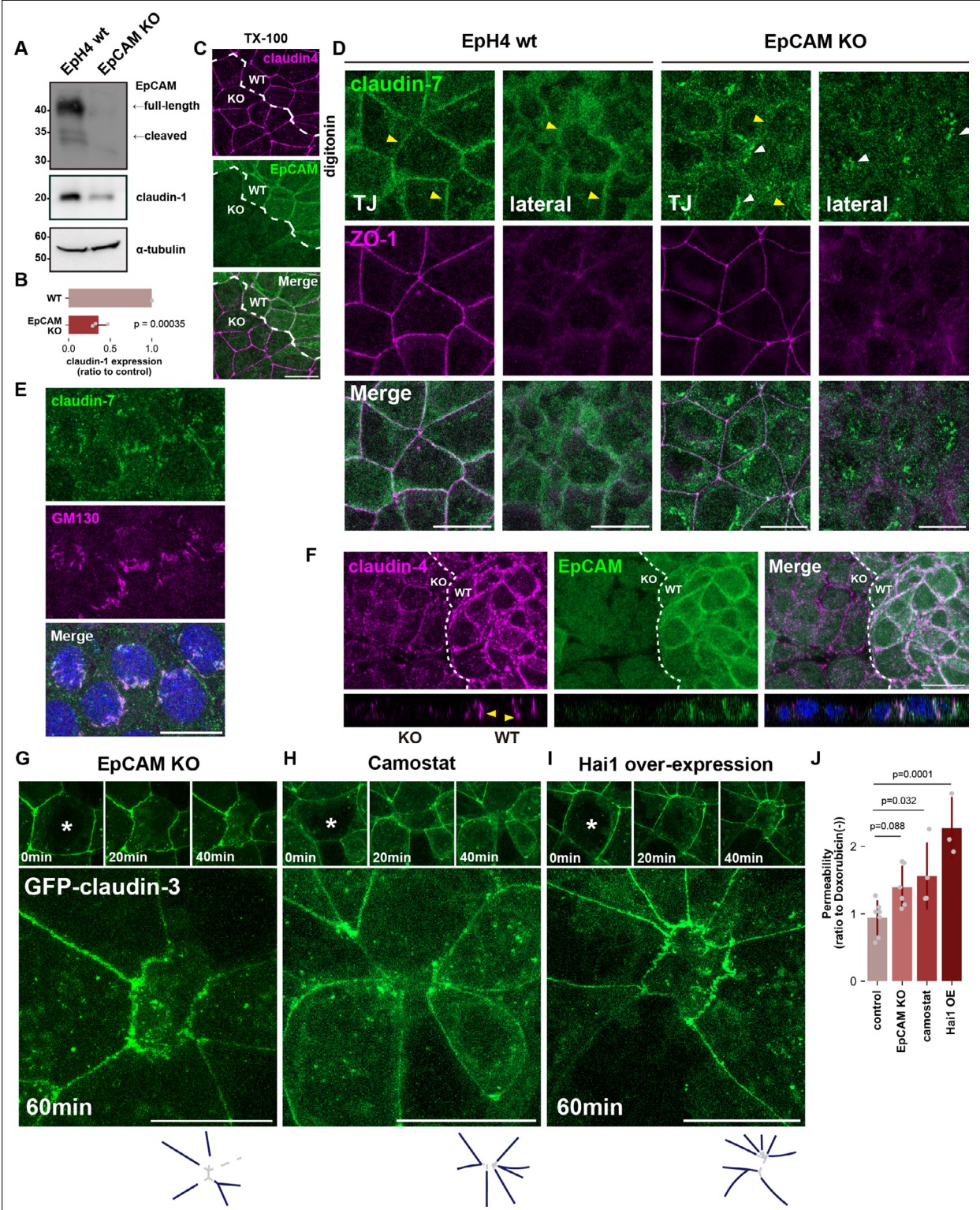

**Figure 8.** Signaling through RhoA-Matriptase-EpCAM is critical for the formation of de novo tight junction (TJ) following apical extrusion. (**A** and **B**) Whole-cell lysates of wild-type (WT) and EpCAM KO EpH4 cells were immunoblotted with the indicated antibodies. Molecular weight measurements are in kDa. The expression level of claudin-1 normalized to α-tubulin was quantified in (**B**) (N=3; error bar: ± SD; Student's t test). (**C**) WT and EpCAM KO EpH4 cells were co-cultured, fixed, and then stained with anti-EpCAM pAb (green) and anti-claudin-4 mAb (magenta). Dotted line overlays the border between WT and EpCAM KO cells. Scale bar: 20 μm. (**D**) WT or EpCAM KO EpH4 cells were cultured, fixed, permeabilized with digitonin, and then stained with anti-claudin-7 pAb (green) and anti-ZO-1 mAb (magenta). Yellow arrowheads indicate claudin-7 at the lateral membrane, while white

*Figure 8 continued on next page*

*Figure 8 continued*

arrowheads indicate the intercellular accumulation of claudin-7. Scale bar: 20 µm. (**E**) EpCAM KO EpH4 cells permeabilized with digitonin were stained with anti-claudin-7 pAb (green) and anti-GM130 mAb (magenta; Golgi marker). Scale bar: 20 µm. (**F**) WT and EpH4 KO EpH4 cells were co-cultured, incubated in acid buffer (pH 6.0) for 20 min, fixed, and then stained with anti-claudin-4 mAb (green) and anti-EpCAM pAb (magenta). Yellow arrowheads indicate the formation of ectopic TJ strands on the basolateral membrane. Scale bar: 20 µm. (**G**, **H**, and **I**). Live cell imagings of EpCAM KO EpH4 cells expressing GFP-claudin-3 (**G**), EpH4 cells expressing GFP-claudin-3 treated with Camostat (**H**), or EpH4 expressing both Hai1-mScarlet and GFP-claudin-3 (**I**) after laser irradiation. Cell marked by an asterisk was wounded at time zero. The right diagram depicts the condition of TJs 60 min after laser irradiation. The navy color represents pre-existing TJs, and the dashed lines indicate the absence of newly formed TJs between neighboring cells. Scale bar = 20 µm. (See also **Videos 4–6**). (**J**) Bar graph showing the paracellular flux of 70 kDa FITC-dextran tracer molecule at 9 hr post-doxorubicin treatment (4 µM) (control: N=7, EpCAM KO: N=6, Camostat: N=4, Hai1 over-expression (OE): N=3; error bar: ± SD; Dunnett's test).

The online version of this article includes the following source data for figure 8:

**Source data 1.** Full blot original image for the Western blot shown in *Figure 8A*.

**Source data 2.** Full blot data for the Western blot shown in *Figure 8A*.

further investigate how the interaction of EpCAM/TROP2 and claudin at the trans-Golgi networks helps the intracellular transport of claudin.

# Materials and methods

## Antibodies and other reagents

The following primary antibodies were used for immunofluorescence microscopy and immuno-blotting: rabbit anti-claudin-3 pAb (34–1700; Thermo Fisher Scientific); mouse anti-claudin-4 mAb (32–9400; Thermo Fisher Scientific); and rabbit anti-claudin-7 pAb (34–9100; Thermo Fisher Scientific); rabbit anti-α-catenin pAb (C2081; Sigma Aldrich); rabbit anti-claudin-1 pAb (SAB4200534; Sigma Aldrich); rat anti-E-cadherin mAb (ECCD2; Takara Bio); rabbit anti-RhoA mAb (2117; Cell Signaling Technology); rabbit anti-phospho-Myosin Light Chain 2 (Thr18/Thr19) mAb (95777; Cell Signaling Technology); rabbit anti-TROP2 mAb (ab214488; abcam); rabbit anti-EpCAM pAb (ab71916; abcam); mouse anti-GM130 mAb (610822; BD). Mouse anti–α-tubulin (12G10) mAb, rat anti-occludin (Moc37) mAb, mouse anti–ZO-1 (T8-754) mAb, and mouse myosin IIB (CMII-23) were produced in-house. Mouse matriptase mAbs, M24 and M69, were generous gifts from Dr. Chen-yong Lin, Georgetown University Medical Center, Washington DC. Rat anti–α-catenin mAb (α18) was a generous gift from Dr. A. Nagafuchi, Nara Medical University, Nara, Japan.

Secondary antibodies were as follows: Cy2-conjugated donkey anti-rat IgG antibody (712-225-150), anti-mouse IgG antibody (715-225-151), and anti-rabbit-IgG antibody (711-225-152); Cy3-conjugated donkey anti-rat IgG antibody (712-165-153), anti-mouse IgG antibody (715-165-150), and anti-rabbit IgG antibody (711-165-152); Cy5-conjugated donkey anti-rabbit IgG antibody (711-175-152; Jackson ImmunoResearch) and HRP-conjugated anti-rat IgG antibody (HAF005; R&D Systems), anti-mouse-IgG antibody (A90-516P; Bethyl Laboratories), and anti-rabbit IgG antibody (4030–05; Southern Biotech).

F-actin was visualized with Alexa Fluor 647–phalloidin (A22287; Thermo Fisher Scientific). Nucleus was visualized with DAPI (049–18801; Wako Pure Chemical Industries).

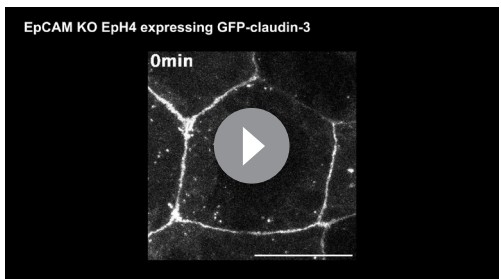

**Video 4.** EpCAM KO EpH4 cells expressing GFP-claudin-3 in cells neighboring laser-wounded cells were imaged. Frames were taken every 30 s. (Scale bar, 20 µm.)

https://elifesciences.org/articles/102794/figures#video4

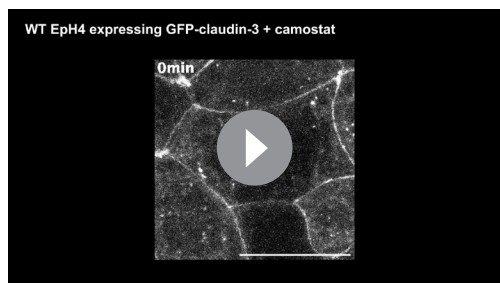

**Video 5.** EpH4 cells expressing GFP-claudin-3 in cells neighboring laser-wounded were imaged. Cells were pre-treated with Camostat (matriptase inhibitor). Frames were taken every 5 min. (Scale bar, 20 µm.)

https://elifesciences.org/articles/102794/figures#video5

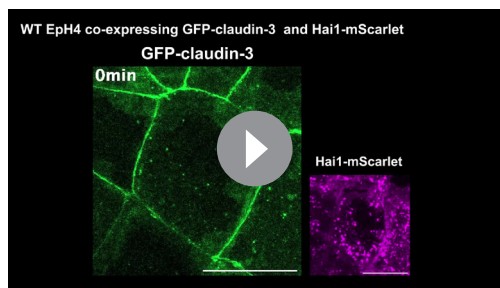

**Video 6.** EpH4 cells co-expressing GFP-claudin-3 and Hai1-mScarlet in cells neighboring laser-wounded were imaged. Frames were taken every 5 min. (Scale bar, 20 µm.)

https://elifesciences.org/articles/102794/figures#video6

Chemicals and recombinant proteins were obtained as follows: H-1152 (555550; Sigma Aldrich); Y27632 (08945–84; Nacalai Tesque); Doxorubicin Hydrochloride (040–21521; Wako); SP600125 (197–16591; Wako); Rho activator II (CN03-A; Cytoskeleton Inc); Camostat Mesylate (C2977; Tokyo Chemical Industry Co., Ltd.).

## Cell culture

EpH4 cells, HaCaT cells, and HEK293 cells were grown in DMEM supplemented with 10% (vol/vol) FCS. All cell cultures were maintained in 5% $CO_2$ atmosphere at 37 °C. EpH4 cells were kindly provided by Dr. Reichmann (Institut Suisse de Recherches, Lausanne, Switzerland), HaCaT cells by Dr. Kiyoshima (Graduate School of Dental Science, Kyushu University), and HEK293 cells were obtained from the ATCC. HaCaT cells and HEK293 cells were authenticated by short tandem repeat (STR) profiling using the GenePrint10 System (Promega). For EpH4 cells, amplification of mouse-specific genomic sequences was confirmed. All cell lines were routinely tested and confirmed to be negative for mycoplasma contamination.

For live imaging, cells were cultured on 35 mm glass-base dish for 48 hr before observation. For immunostaining, cells were cultured on 15 mm coverslips. For mix-culture experiments, two types of cells were co-cultured at a ratio of 1:1.

For differentiation of HaCaT cells into granular-like cells, cells were cultured in high-calcium (9.8 mM) medium supplemented with a JNK inhibitor (40 µM) for 24 h. For inhibition of ROCK, cells were treated with H-1152 (final 2 µM) or Y27632 (final 20 µM) for 30 min. For inhibition of serine protease, cells were treated with Camostat (final 10 µM) for 6 hr. For activation of Rho, after serum starved for 24 hr, cells were treated with Rho activator (final 1 µg/ml) for 2 hr. For activation of ROCK, cells were treated with Narciclasine (final 100 nM) for 4 hr. For activation of matriptase, cells were cultured in pH 6.0 citric acid-phosphate buffer for 20 min.

## Plasmids

We established KO cells by using the CRISPR-Cas9 system. Oligonucleotides were phosphorylated, annealed, and cloned into the BsmBI site of pLenti-CRISPR v2 vector according to the Zhang laboratory protocols (F. Zhang, MIT, Cambridge, MA). The target sequences were as follows:

> Mouse EpCAM: 5′- GGGCGATCCAGAACAACGAT –3′
> Human TROP2: 5′- GGCGGCGGTGACCGGCCACA –3′

Other expressing vectors were constructed on the pLenti-CMV backbone.

## Transfection and generation of knockout cell lines

Transfections were performed by using the PEI-max (24765–1; Polysciences Inc). Lentiviruses were produced in HEK293 cells by transfecting the expression vector with the packaging and envelope vectors (psPAX2 and pMD2.G) and harvested by centrifugation. Infections were carried out in low calcium medium with polybrene. Cells remained in infection media for 48 hr, followed by antibiotic selection (400 µg/ml neomycin; 4.5 µg/ml puromycin; 200 µg/ml hygromycin).

To establish knockout cell lines, cells were seeded into 96-well plates after drug selection, and at least 10 clones were collected and examined by western blotting to check for loss of expression of the protein of interest. HaCaT cells expressing GFP-claudin-3 or Hai1-GFP and EpH4 cells expressing GFP-claudin3 and/or Hai1-scarlet were obtained using a cell sorter (SH800; SONY).

## Live imaging

Fluorescence imaging was performed using a confocal microscope (LSM900; Carl Zeiss MicroImaging) equipped with a 63 x/1.40 NA, oil immersion Plan-APO objective with a heated stage set to 37 °C, and cells were cultured in Leibovitz's L-15 Medium (1415064; Thermo Fisher Scientific) supplemented

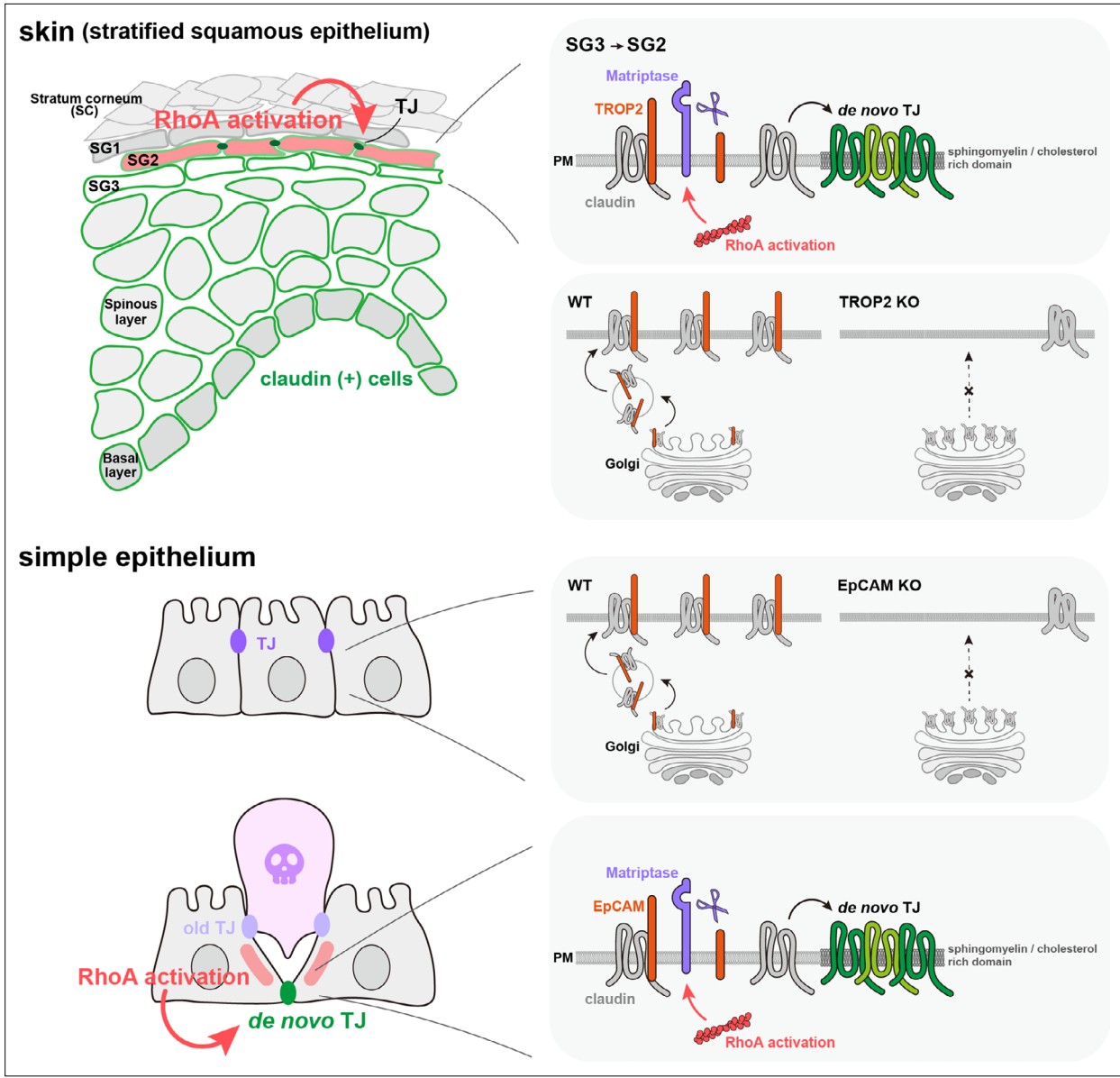

**Figure 9.** Activation of Rho-ROCK pathway triggers de novo tight junction (TJ) formation via disruption of the EpCAM/TROP2-claudin complex. De novo TJ formation associated with apoptotic cell elimination in simple epithelia and with keratinocyte differentiation in stratified epithelia is regulated by a common molecular mechanism. In both types of epithelia, EpCAM or TROP2 is necessary for the intracellular transport of claudin from the Golgi apparatus to the plasma membrane. Claudin, when in complex with EpCAM or TROP2, cannot spontaneously polymerize into TJs at the plasma membrane. However, upon activation of the Rho-ROCK pathway, which activates matriptase, EpCAM or TROP2 is cleaved by matriptase. This cleavage breaks down the EpCAM/TROP-2-claudin complex, enabling claudin to polymerize into TJs. Consequently, rapid TJ formation occurs without de novo transcription or translation.

with 10% (vol/vol) FCS. Images were acquired using Zen 2012 (Carl Zeiss MicroImaging). For induced apoptosis, nuclei of any selected cells were injured by MicroPoint Laser (wavelength = 435 nm, duration <4 ns, laser energy <40 µJ, single pulse, OXFORD Instruments).

## Immunofluorescence microscopy

Cells cultured on coverslips were fixed with 3% formalin prepared in PBS for 15 min at room temperature (RT), permeabilized with 0.4% Triton X-100/PBS for 5 min and blocked with 1% BSA prepared in PBS for 1 hr at RT. In *Figure 8D and E*, cells were fixed with 4% paraformaldehyde (PFA)/PBS for 15 min and permeabilized with 100 µM digitonin/PBS for 10 min at RT. When using anti-matriptase mAbs, cells were fixed with 4% PFA/PBS and permeabilized with 0.4% Triton X-100/PBS. Antibodies

were diluted in the blocking solution. Cells were incubated with primary antibodies for 1 hr at RT and with secondary antibodies for 30 min at RT. Samples were observed at RT with the confocal microscope.

To quantify junctional enrichment of indicated molecule, cells were segmented based on the maximum projection of E-cadherin immunostaining using the Cellpose 2 according to the schematic in *Figure 3—figure supplement 1*; *Pachitariu and Stringer, 2022*. Junction areas were obtained by the 'find edge' and 'dilute' commands in ImageJ. Junctional signal intensities were obtained by the 'Image calculator' command in ImageJ. Finally, junctional enrichment of the indicated molecule was quantified as the signal intensity of the indicated molecule divided by the junction area.

## Mouse ear whole-mount immunofluorescence microscopy

The detachment of the epidermis from the mouse ear and fluorescence immunostaining were performed as previously described (*Kubo et al., 2022*). In brief, after euthanasia of ICR mouse, the ears were immediately cut off at its root and floated in calcium-containing PBS. The dorsal skin was removed, followed by the removal of cartilage from the ventral side skin. The epidermis and dermis were separated after dispase treatment. The separated epidermis was fixed with 4% PFA for 10 min and permeabilized with 1% Triton X-100 for 10 min. After washing, samples were blocked with 1% BSA/ PBS and incubated with primary antibodies overnight at 4 °C and with secondary antibodies for 2 hr at RT. The samples were mounted in Mowiol. Fluorescence imaging was performed using a confocal microscope equipped with a ×40 /1.40 NA, oil immersion Plan-APO objective. All animal experiments were performed in accordance with the Guidelines for Animal Experiments of Kobe University and approved by the Institutional Animal Care and Use Committee (Permit Number: P211104).

## Immunoblotting

Cells were cultured in 30 mm dish or six-well plate and lysed in SDS sample buffer and boiled for 5 min. Lysates were prepared in SDS sample buffer without 2-Mercaptoethanol and were not boiled. Samples were resolved by SDS-PAGE and transferred to nitrocellulose membrane. The membrane was blocked with 5% skim milk/0.1% Tween 20/TBS (TBST) for 1 hr at RT. Indicated proteins were probed by sequential incubation with the primary antibody and HRP-conjugated secondary antibody prepared in 5% BSA/TBST for 1 hr at RT each. After each antibody reaction, nitrocellulose membranes were washed three times with TBST for 10 min. Signals were detected by mixing A (100 mM Tris-HCl [pH 8.5], 0.4 mM p-coumaric acid, 5 mM luminol) and B (100 mM Tris-HCl [pH 8.5], 0.04% $H_2O_2$). Chemiluminescence was captured using the LAS-3000 Imaging system (Fujifilm).

## Cell surface protein biotinylation

Cell surface protein biotinylation was performed according to a previously reported protocol (*Tarradas et al., 2013*). The biotinylation of surface proteins was performed using EZ-Link sulfo-NHS-SS-biotin (21217; Thermo Fisher Scientific). Biotinylated proteins were isolated using High-Capacity Streptavidin Agarose Resin (20361; Thermo Fisher Scientific).

## Paracellular tracer flux measurement

$10^5$ EpH4 cells were seeded on a 12-mm-diameter cell culture insert (83.3931.040; SARSTEDT) and cultured with daily medium changes. After 6 days, cells were basally treated with 4 µM Doxorubicin Hydrochloride for 9 hr, and 70 kDa FITC-dextran (46945; Sigma Aldrich) was added to the medium in the apical compartment at a concentration of 1 mg/ml. Medium was collected from the basal compartment after 1 hr, and the FITC signal was measured with a fluorometer.

## Active RhoA pulldown assay

RhoA activation was assessed using GST-tagged rhotekin RBD domain (GST-RBD), which binds to the GTP-bound form of RhoA. Undifferentiated or differentiated HaCaT cells were lysed using lysis buffer (20 mM HEPES (pH 7.5), 150 mM NaCl, 5 mM MgCl2, 1% Triton X, 1 mM DTT, 100 µM APMSF, 10 µg/ ml aprotinin, 10 µg/ml leupeptin), and after centrifugation, the supernatant was aliquoted as input and the remainder incubated with GST-RBD bound to glutathione Sepharose beads at 4 °C for 2 hr. After washing with lysis buffer, pulldown samples were eluted with SDS sample buffer. Immunoblotting was performed using an anti-RhoA mAb, and the band intensities of RhoA in both input and pulldown

samples were quantified using Fiji/ImageJ. RhoA activity was calculated by dividing the intensity of activated RhoA (pulldown) by the intensity of input, with undifferentiated cells set to 1.00.

## Quantification and statistical analysis

Error bars indicate 95% confidence intervals. The number and statistical details of independent experiments are indicated in figure legend. Each point on the graph represents an individual biological replicate. Unless otherwise specified, microcopy images are representatives of at least three independent experiments. The microscope images of the mouse epidermis are representative ones sourced from a single mouse. Statistical analysis was performed in Python 3.11.3 using scipy 1.10.1, statsmodels 0.14.0, and scikit_posthocs 0.9.0 packages. Graphs were created using matplotlib 3.7.1 and seaborn 0.12.2. The number and statistical details of independent experiments are indicated in figure legend. Error bar indicates SD.

## Acknowledgements

We thank all members of the Ikenouchi laboratory (Department of Biochemistry, Faculty of Medical Sciences, Kyushu University) and Prof. Kyoko Shirakabe (Ritsumeikan Univ) for helpful discussions. This work was supported by JSPS KAKENHI (JP22H02618, JP23K18141, JP25H01325 and JP25H00994 [JI], JP23KJ1689 [YC]), JST-FOREST (JPMJFR204L) (JI), and Research Grants from Takeda Science Foundation and KOSÉ Cosmetology Research Foundation (JI).

## Additional information

### Funding

| Funder | Grant reference number | Author |
| --- | --- | --- |
| Japan Society for the Promotion of Science | JP22H02618 | Junichi Ikenouchi |
| Japan Society for the Promotion of Science | JP23K18141 | Junichi Ikenouchi |
| Japan Society for the Promotion of Science | JP25H01325 | Junichi Ikenouchi |
| Japan Society for the Promotion of Science | JP25H00994 | Junichi Ikenouchi |
| Japan Society for the Promotion of Science | JP23KJ1689 | Yuma Cho |
| Japan Science and Technology Agency | JPMJFR204L | Junichi Ikenouchi |
| Takeda Science Foundation | | Junichi Ikenouchi |
| KOSÉ Cosmetology Research Foundation | | Junichi Ikenouchi |

The funders had no role in study design, data collection and interpretation, or the decision to submit the work for publication.

### Author contributions

Yuma Cho, Conceptualization, Resources, Data curation, Formal analysis, Validation, Investigation, Visualization, Writing – original draft; Akari Taniguchi, Resources, Data curation, Formal analysis, Investigation; Akiharu Kubo, Resources, Data curation, Investigation, Visualization, Methodology; Junichi Ikenouchi, Conceptualization, Supervision, Writing – original draft, Project administration, Writing – review and editing

### Author ORCIDs

Yuma Cho ⓘ https://orcid.org/0009-0008-6847-8947

Akiharu Kubo http://orcid.org/0000-0003-0902-3586
Junichi Ikenouchi https://orcid.org/0000-0002-2936-3548

### Ethics

All animal experiments were performed in accordance with the Guidelines for Animal Experiments of Kobe University and approved by the Institutional Animal Care and Use Committee (Permit Number: P211104).

Reviewer #1 (Public review): https://doi.org/10.7554/eLife.102794.3.sa1
Reviewer #2 (Public review): https://doi.org/10.7554/eLife.102794.3.sa2
Author response https://doi.org/10.7554/eLife.102794.3.sa3

---

# Additional files

### Supplementary files

MDAR checklist

### Data availability

All data generated or analysed during this study are included in the manuscript and supporting files.

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
