## [Editor Report · eLife Assessment]

This paper identifies a crucial step in the regulation of tight junction formation by identifying Rho-ROCK activity-dependent activation of the serine protease Matriptase, making Claudins available for tight junction formation. The reviewers were satisfied with the revisions and found the work **important** and the approach **convincing**.

---

## [Referee Report · Reviewer #1 (Public review)]

Summary:

The manuscript by Cho and colleagues investigates de novo tight junction formation during the differentiation of immortalized human HaCaT keratinocytes to granular-like cells, as well as during epithelial remodeling that occurs upon the apoptotic of individual cells in confluent monolayers of the representative epithelial cell line EpH4. The authors demonstrate the involvement of Rho-ROCK with well-conducted experiments and convincing images. Moreover, they unravel the underlying molecular mechanism, with Rho-ROCK activity activating the transmembrane serine protease Matriptase, which in turn leads to the cleavage of EpCAM and TROP2, respectively, releasing Claudins from EpCAM/TROP2/Claudin complexes at the cell membrane to become available for polymerization and de novo tight junction formation. These functional studies in two different cell culture systems are complemented by localization studies of the according proteins in the stratified mouse epidermis in vivo.

In total, these are new and very intriguing and interesting findings that add important new insights into the molecular mechanisms of tight junction formation, identifying Matriptase as the "missing link" in the cascade of formerly described regulators. The involvement of TROP2/EpCAM/Claudin has been reported recently (Szabo et al., Biol. Open 2022; Bugge lab), and Matriptase had been formerly described to be required for tight junction formation as well, again from the Bugge lab. Yet, the functional correlation / epistasis between them, and their relation to Rho signaling, had not been known thus far.

Strengths:

Convincing functional studies in two different cell culture systems, complemented by supporting protein localization studies in vivo. The manuscript is clearly written and most data are convincingly demonstrated, with beautiful images and movies.

Weaknesses:

The previously described weaknesses have been fully wiped out during the revisions.

---

## [Referee Report · Reviewer #2 (Public review)]

Summary:

In this manuscript the authors investigate how epithelia maintain intercellular barrier function despite and during cellular rearrangements upon e.g. apoptotic extrusion in simple epithelia or regenerative turnover in stratified epithelia like this epidermis. A fundamental question in epithelial biology. Previous literature has shown that Rho mediated local regulation of actomyosin is essential not only for cellular rearrangement itself but also directly controls tight junction barrier function. The molecular mechanics however remained unclear. Here the authors use extensive fluorescence imaging of fixed and live cells together with genetic and drug mediated interference to show that Rho activation is required and sufficient to form de novo tight junctional strands at intercellular contacts in epidermal keratinocytes (HaCat) and mammary epithelial cells. After having confirmed previous literature they then show that Rho activation activates the transmembrane protease matriptase which cleaves EpCAM and TROP2, two claudin binding transmembrane proteins, to release claudins and enable claudin strand formation and therefore tight junction barrier function.

Strengths:

The presented mechanism is shown to be relevant for epithelial barriers being conserved in simple and stratifying epithelial cells and mainly differs due to tissue specific expression of EpCAM and TROP2. The authors present carefull state of the art imaging and logical experiments that convincingly support the statements and conclusion. The manuscript is well written and easy to follow.

Weaknesses:

Whereas the in vitro evidence of the presented mechanism is strongly supported by the data, the in vivo confirmation is mostly based on the predicted distribution of TROP2. Whereas the causality of Rho mediated matriptase activation has been nicely demonstrated it remains unclear how Rho activates matriptase.

---

## [Author Response]

The following is the authors’ response to the original reviews.

**Public Reviews:**

**Reviewer #1 (Public review):**
Summary:The manuscript "Rho-ROCK liberates sequestered claudin for rapid de novo tight junction formation" by Cho and colleagues investigates de novo tight junction formation during the differentiation of immortalized human HaCaT keratinocytes to granular-like cells, as well as during epithelial remodeling that occurs upon the apoptotic of individual cells in confluent monolayers of the representative epithelial cell line EpH4. The authors demonstrate the involvement of Rho-ROCK with well-conducted experiments and convincing images. Moreover, they unravel the underlying molecular mechanism, with Rho-ROCK activity activating the transmembrane serine protease Matriptase, which in turn leads to the cleavage of EpCAM and TROP2, respectively, releasing Claudins from EpCAM/TROP2/Claudin complexes at the cell membrane to become available for polymerization and de novo tight junction formation. These functional studies in the two different cell culture systems are complemented by localization studies of the according proteins in the stratified mouse epidermis in vivo.In total, these are new and very intriguing and interesting findings that add important new insights into the molecular mechanisms of tight junction formation, identifying Matriptase as the "missing link" in the cascade of formerly described regulators. The involvement of TROP2/EpCAM/Claudin has been reported recently (Szabo et al., Biol. Open 2022; Bugge lab), and Matriptase had been formerly described to be required for in tight junction formation as well, again from the Bugge lab. Yet, the functional correlation/epistasis between them, and their relation to Rho signaling, had not been known thus far.However, experiments addressing the role of Matriptase require a little more work.Strengths:Convincing functional studies in two different cell culture systems, complemented by supporting protein localization studies in vivo. The manuscript is clearly written and most data are convincingly demonstrated, with beautiful images and movies.Weaknesses:The central finding that Rho signaling leads to increased Matriptase activity needs to be more rigorously demonstrated (e.g. western blot specifically detecting the activated version or distinguishing between the full-length/inactive and processed/active version).

First, we thank the reviewer for their fair evaluation of our manuscript and for providing constructive feedback. Regarding the detection of matriptase activation—which Reviewer 1 identified as a weakness—we fully agree that direct validation is crucial. Therefore, in this revision we have carried out additional experiments using the M69 antibody, which specifically recognizes the activated form of matriptase. Details of these new experiments are provided in our point-by-point responses below.

**Reviewer #2 (Public review):**
Summary:In this manuscript, the authors investigate how epithelia maintain intercellular barrier function despite and during cellular rearrangements upon e.g. apoptotic extrusion in simple epithelia or regenerative turnover in stratified epithelia like this epidermis. A fundamental question in epithelial biology. Previous literature has shown that Rho-mediated local regulation of actomyosin is essential not only for cellular rearrangement itself but also for directly controlling tight junction barrier function. The molecular mechanics however remained unclear. Here the authors use extensive fluorescent imaging of fixed and live cells together with genetic and drug-mediated interference to show that Rho activation is required and sufficient to form novo tight junctional strands at intercellular contacts in epidermal keratinocytes (HaCat) and mammary epithelial cells. After having confirmed previous literature they then show that Rho activation activates the transmembrane protease Matriptase which cleaves EpCAM and TROP2, two claudin-binding transmembrane proteins, to release claudins and enable claudin strand formation and therefore tight junction barrier function.Strengths:The presented mechanism is shown to be relevant for epithelial barriers being conserved in simple and stratifying epithelial cells and mainly differs due to tissue-specific expression of EpCAM and TROP2. The authors present careful state-of-the-art imaging and logical experiments that convincingly support the statements and conclusion. The manuscript is well-written and easy to follow.Weaknesses:Whereas the in vitro evidence of the presented mechanism is strongly supported by the data, the in vivo confirmation is mostly based on the predicted distribution of TROP2. Whereas the causality of Rho-mediated Matriptase activation has been nicely demonstrated it remains unclear how Rho activates Matriptase.

Thank you for your valuable feedback on our manuscript. As Reviewer 2 points out, the precise mechanism by which the Rho/ROCK pathway activates matriptase remains unclear. We have discussed the possible molecular mechanisms in the Discussion section. Elucidating the detailed mechanism of matriptase activation will be the focus of our future work.

**Recommendations for the authors:**

**Reviewer #1 (Recommendations for the authors):**
Comment 1-1 - Matriptase activation by Rho: The authors show activation of Matriptase in western blots by the simple reduction of (full-length?) protein level in Figures 5 and 7. Most publications however show activated Matriptase either by antibodies detecting specifically the active form (including the publication referenced in this manuscript), or the appearance of the activated form next to the inactive form (based on different molecular weights). Therefore, it is not completely clear whether the treatment with Rho activators (Figure 5) results in an overall decrease of Matriptase, or really in an increase in the activated form. Therefore, the authors should show the actual increase of the active form. As a control, the impact of camostat treatment and overexpression of Hai1 on the active form of Matriptase could be included. It also should be indicated in the figure legend how long cells had been treated with the drugs before being subjected to lysis. Moreover, the western blots need to be quantified.

We performed a more rigorous analysis using the M69 antibody, which specifically recognizes the activated form of matriptase and has been widely used in previous studies（e.g. Benaud et al., 2001; Hung et al., 2004; Wang et al., 2009）. We likewise confirmed a significant increase in M69 signals by both western blotting and immunostaining from samples in which matriptase was activated by acid medium treatment (Figure 5A). Crucially, we also observed matriptase activation with the M69 antibody both in Rho/ROCK activator-treated cells (Figure 5A) and in differentiated granular-layer-like cells (Figures 7A and 7D). These findings strongly support the conclusion that matriptase is activated downstream of the Rho/ROCK pathway.

Comment 1-2 - Based on their results, the authors conclude that Matriptase cleaves TROP2 in the SG2 layer of the epidermis, which is a little contradictory to former studies, which have shown Matriptase to be most prominently expressed and active in the basal layer and only little in the spinous layer (e.g Chen et al., Matriptase regulates proliferation and early, but not terminal, differentiation of human keratinocytes. J Invest Dermatol.2013). In this light, one could also argue that inhibiting Matriptase "simply" reduces epidermal differentiation. Can other differentiation markers be tested to rule that the effects on tight junctions are secondary consequences of interferences with earlier / more global steps of keratinocyte differentiation?

As the reviewer noted, previous studies have demonstrated that matriptase is essential for keratinocyte differentiation, and that it cleaves substrates beyond EpCAM and TROP2—any of which could potentially influence the differentiation process. To test this possibility, we chose to monitor maturation of adherens junction (AJ) as an indicator of keratinocyte differentiation into granular-layer cells. Prior work has shown that during differentiation into granular-layer cells, AJs develop and experience increased intercellular mechanical tension, and that this rise in mechanical tension at AJs is critical for subsequent TJ formation (Rübsam et al., 2017). To assess AJ tension, we stained with the α-18 monoclonal antibody, which specifically recognizes the tension-dependent conformational change of α-catenin, a core AJ component. In control cells, differentiation into granular-layer like cells led to a marked increase in α-18 signal at cell–cell adhesion sites. Importantly, when HaCaT cells were treated with Camostat to inhibit matriptase and then induced to differentiate, we observed an equivalent increase in α-18 signal at AJs (Figure 7F). However, we did not detect claudin enrichment at cell-cell contacts under these conditions (Figures 7F and 7H). These results suggest that matriptase inhibition does not impair AJ maturation during granular-layer differentiation, but does profoundly disrupt TJ formation. While we cannot rule out the possibility that matriptase acts more broadly from these results, we judged that a comprehensive substrate survey lies outside the scope of the present manuscript.

Comment 1-3 - In addition, as in Figure 5, full-length levels of Matriptase in Figure 7A need to be complemented by the active version to demonstrate more convincingly that TROP2 processing coincides with (and is most likely caused by) increased Matriptase activation. In the quantification in 7B, levels actually go up again after 2 and 4 hours. How is that explained, and what would this mean with respect to tight junction formation seen at 24 h of differentiation? The TROP2 cleavage shown in Figure 7A should be quantified.

This comment is related to Comment 1-1. Using the M69 antibody, which specifically recognizes the activated matriptase, we directly demonstrated that matriptase activation occurs during the differentiation of granular layer-like cells (Figures 7A and 7D). Furthermore, we performed quantitative analysis of TROP2 cleavage and found that, compared with undifferentiated cells, differentiation into granular-layer like cells was accompanied by an increase in the cleaved TROP2 fragments (Figures 7A and 7B).

Minor points:Comment 1-4 - Figure 1B and C: Including orthogonal views would be a nice add-on to appreciate the findings.

In the revised version, we have added the corresponding orthogonal views to Figure 1B and Figure 1C.

Comment 1-5 - Figure 2D: last row: indication of orthogonal view.

We stated that the bottom panels are orthogonal views in the figure legend of Figure 2D.

Comment 1-6 - Figure 3A: quantification is missing. GST-Rhotekin assay is missing in methods.

In the revised manuscript, we have added quantitative analysis for Figure 3A. We have also supplemented the Materials and Methods section with detailed information on the GST–Rhotekin assay used to quantify levels of active RhoA.

Comment 1-7 - Figure 4H: quantification of the Western blot is missing.

In the revised manuscript, we have added quantitative analysis for Figure 4H as Figure 4I.

Comment 1-8 - Figure 5 and 6: Quantifications of Western blots are missing.

In the revised manuscript, we have added quantitative analyses for Figure 5D as Figure 5F and for Figure 6A as Figure 6B.

Comment 1-9 - Figure 7C: quantification of the Western blot is missing.

Figure 7C does not present western blotting data. For the other western blotting results, we have added quantitative analyses as suggested by Reviewer 1.

Comment 1-10 - Figure 8I: Including Hai1 overexpression would be good for a complete picture.

Following Reviewer 1’s suggestion, we have added staining data for Hai1-overexpressing cells to Figure 8J.

Comment 1-11 - Line 377: The authors say they found Matriptase always present in lateral membranes. I did not find evidence for this in the manuscript.

Previous studies have demonstrated that in polarized epithelial cells, matriptase is localized to the basolateral membrane below TJs (Buzza et al., 2010; Wang et al., 2009). We also found that matriptase consistently localizes to the basolateral membrane but more crucially that it becomes activated there during differentiation into granular layer cells. We added these new data as Figures 7C-7E in the revised manuscript. These findings suggest that matriptase activation occurs without a change in its subcellular localization.

Comment 1-12 - Line 381: should most likely say: and ADAM17 but it is not known whether...

We corrected the sentence in the revised manuscript.

**Reviewer #2 (Recommendations for the authors):**
The authors have added a significant number of quantifications verifying their observations, which was a major comment in a previous version of the manuscript and thus I have only a few minor comments which should be addressed.Comment 2-1 - It is not required to have scale bars in every image of a panel if the same scale is used.

Unnecessary scale bars were removed. Specifically, scale bars were removed from Figure 1B, 1C, 1F, 8F, 8G, and 8H.

Comment 2-2 - Throughout all figures: Please state for non-quantified images whether this is a representative example and for how many technical or biological repeats this is representative. Also for "N" number, state what the N stands for and if this is what the dots in the graph represent. Are these the number of junctions or technical, experimental or biological repeats?

In the revised manuscript, we have added the number of independent experiments and corresponding “N” values to the Quantification and Statistical Analysis subsection of the Materials and Methods.

Comment 2-3 - Some Zooms have a scale bar (6d), and some do not (e.g. 5b).

The scale bar was removed from the magnified image in Figure 6D.